# The differentiation and integration of the hippocampal dorsoventral axis are controlled by two nuclear receptor genes

Xiong Yang[1†], Rong Wan[1†], Zhiwen Liu[2,3], Su Feng[2,3], Jiaxin Yang[1], Naihe Jing[2,3]*, Ke Tang[1]*

[1]Precise Genome Engineering Center, School of Life Sciences, Guangzhou University, Guangzhou, China; [2]Guangzhou Laboratory/Bioland Laboratory, Guangzhou, China; [3]CAS Key Laboratory of Regenerative Biology, Guangdong Provincial Key Laboratory of Stem Cell and Regenerative Medicine, Guangdong Institutes of Biomedicine and Health, Chinese Academy of Sciences, Guangzhou, China

**\*For correspondence:**
njing@sibcb.ac.cn (NJ);
ktang.sc@gmail.com (KT)

[†]These authors contributed equally to this work

**Competing interest:** The authors declare that no competing interests exist.

**Abstract** The hippocampus executes crucial functions from declarative memory to adaptive behaviors associated with cognition and emotion. However, the mechanisms of how morphogenesis and functions along the hippocampal dorsoventral axis are differentiated and integrated are still largely unclear. Here, we show that *Nr2f1* and *Nr2f2* genes are distinctively expressed in the dorsal and ventral hippocampus, respectively. The loss of *Nr2f2* results in ectopic CA1/CA3 domains in the ventral hippocampus. The deficiency of *Nr2f1* leads to the failed specification of dorsal CA1, among which there are place cells. The deletion of both *Nr2f* genes causes almost agenesis of the hippocampus with abnormalities of trisynaptic circuit and adult neurogenesis. Moreover, *Nr2f1/2* may cooperate to guarantee appropriate morphogenesis and function of the hippocampus by regulating the *Lhx5-Lhx2* axis. Our findings revealed a novel mechanism that *Nr2f1* and *Nr2f2* converge to govern the differentiation and integration of distinct characteristics of the hippocampus in mice.

## eLife assessment

This is an **important** study demonstrating distinct roles for the nuclear receptor genes COUP-TFI and COUP-TFII in hippocampal development. The strength of evidence is **compelling**, using rigorous state-of-the-art methods to demonstrate functional redundancy of these genes in regulating the Lhx2/Lhx5 axis. The major strengths of the study are the dramatic morphogenic phenotypes, and the resultant altered gene networks. These findings have theoretical or practical implications beyond a single field, and will be of interest to geneticists, developmental neurobiologists and chromatin biologists among others.

## Introduction

Memory, including declarative and nondeclarative memory, unifies our mental world to ensure the quality of life for people of all ages, from newborns to elderly individuals (*Eichenbaum and Cohen, 2014*; *Kandel et al., 2014*). The pioneering studies of Milner and her colleagues revealed that the hippocampus is required for declarative memory but not nondeclarative memory (*Penfield and Milner, 1958*; *Scoville and Milner, 1957*). The discovery of activity-dependent long-term potentiation and place cells provides the neurophysiological basis of hippocampal function (*Bliss and Gardner-Medwin, 1973*; *O'Keefe and Dostrovsky, 1971*). The rodent hippocampus can be divided into the dorsal and ventral domains, corresponding to the posterior and anterior hippocampus in humans,

respectively. In recent decades, numerous studies have supported the Moser theory that the hippocampus is a heterogeneous structure with distinct characteristics of gene expression, connectivity, and functions along its dorsoventral axis (*Bast, 2007*; *Fanselow and Dong, 2010*; *Moser and Moser, 1998*; *Strange et al., 2014*). The dorsal hippocampus, which connects and shares similar gene expression with the neocortex (*Fanselow and Dong, 2010*), serves the 'cold' cognitive function associated with declarative memory and spatial navigation. The ventral hippocampus, which connects and generates similar gene expression with the amygdala and hypothalamus (*Cenquizca and Swanson, 2007*; *Kishi et al., 2000*; *Pitkänen et al., 2000*), corresponds to the 'hot' affective states related to emotion and anxiety (*Fanselow and Dong, 2010*; *Tyng et al., 2017*). Nonetheless, to date, the molecular and cellular mechanisms by which the morphogenesis, connectivity, and functions along the dorsoventral axis of the hippocampus are differentiated and integrated are largely unknown.

The hippocampus, a medial temporal lobe structure in the adult rodent forebrain, originates from the medial pallium (MP) in the medial line of the early dorsal telencephalon. The cortical hem (CH), which is located ventrally to the MP, functions as an organizer for hippocampal development (*Hébert and Fishell, 2008*; *Schuurmans and Guillemot, 2002*). It has been demonstrated that both extrinsic signals, such as WNTs and BMPs, and intrinsic factors, including EMX1, EMX2, LEF1, LHX2, and LHX5, are involved in the regulation of early morphogenesis of the hippocampus. As the earliest *Wnt* gene to be exclusively expressed in the cortical hem, *Wnt3a* is required for the genesis of the hippocampus (*Lee et al., 2000*); in addition, *Lef1* is downstream of Wnt signaling, and the hippocampus is completely absent in *Lef1*$^{neo/neo}$ null mutant mice (*Galceran et al., 2000*). Wnt signaling is essential for early development of the hippocampus. *Emx1* and *Emx2* are mouse homologs of *Drosophila empty spiracles* (*Simeone et al., 1992*). Interestingly, the dorsal hippocampus is smaller in an *Emx1* null mutant (*Yoshida et al., 1997*), while *Emx2* is required for the growth of the hippocampus but not for the specification of hippocampal lineages (*Tole et al., 2000*). Moreover, *Lhx5*, which encodes a LIM homeobox transcription factor and is specifically expressed in the hippocampal primordium, is necessary for the formation of the hippocampus (*Zhao et al., 1999*). *Lhx2*, encoding another LIM homeobox transcription factor, is required for the development of both the hippocampus and neocortex (*Mangale et al., 2008*; *Monuki et al., 2001*; *Porter et al., 1997*). Intriguingly, deficiency of either *Lhx5* or *Lhx2* results in agenesis of the hippocampus, and more particularly, these genes inhibit each other (*Hébert and Fishell, 2008*; *Mangale et al., 2008*; *Roy et al., 2014*; *Zhao et al., 1999*), indicating that the *Lhx5* and *Lhx2* genes may generate an essential regulatory axis to ensure the appropriate hippocampal development. Nevertheless, whether there are other intrinsic genes that participate in the regulation of morphogenesis and function of the hippocampus has not been fully elucidated.

*Nr2f* genes, including *Nr2f1* and *Nr2f2*, encode two transcription factor proteins belonging to the nuclear receptor superfamily (*Yang et al., 2017*). Mutations of *Nr2f1* are highly related to neurodevelopmental disorders (NDDs), such as intellectual disability (ID) and autism spectrum disorders (ASD) (*Bertacchi et al., 2020*; *Bosch et al., 2014*; *Contesse et al., 2019*), and mutations of the *Nr2f2* gene are associated with congenital heart defects (CHDs) (*Al Turki et al., 2014*). By using animal models, our studies and others have demonstrated that *Nr2f* genes participate in the regulation of the development of the central nervous system (*Zhang et al., 2020*). The *Nr2f1* plays an essential role in the differentiation of cortical excitatory projection neurons and inhibitory interneurons, the development of the dorsal hippocampus, and cortical arealization (*Armentano et al., 2007*; *Bertacchi et al., 2020*; *Del Pino et al., 2020*; *Feng et al., 2021*; *Flore et al., 2017*; *Lodato et al., 2011*; *Zhou et al., 1999*; *Zhou et al., 2001*). *Nr2f2* plays a vital role in the development of the amygdala, hypothalamus, and cerebellum (*Feng et al., 2017*; *Kim et al., 2009*; *Tang et al., 2012*). Nevertheless, whether and how *Nr2f1* and/or *Nr2f2* genes regulate the differentiation and integration of hippocampal morphogenesis, connectivity, and function is still largely unclear.

Here, our data show that *Nr2f1* and *Nr2f2* genes are differentially expressed along the dorsoventral axis of the postnatal hippocampus. The loss of *Nr2f2* results in ectopic CA1 and CA3 domains in the ventral hippocampus. In addition, the deficiency of *Nr2f1* leads to not only dysplasia of the dorsal hippocampus but also failed specification and differentiation of the dorsal CA1 pyramidal neuron lineage. Furthermore, the deletion of both genes in the RX$^{Cre/+}$; *Nr2f1*$^{F/F}$; *Nr2f2*$^{F/F}$ double-mutant mouse causes almost agenesis of the hippocampus, accompanied by compromised specification of the CA1, CA3, and dentate gyrus (DG) domains. The components of the trisynaptic circuit are abnormal in the corresponding single-gene or double-gene mutant model. Moreover, *Nr2f* genes

may cooperate to guarantee the appropriate morphogenesis and function of the hippocampus by regulating the *Lhx5-Lhx2* axis.

## Results

### Differential expression profiles of *Nr2f1* and *Nr2f2* genes along the dorsoventral axis in the developing and postnatal hippocampus

To investigate the functions of the *Nr2f1* and *Nr2f2* genes in the hippocampus, immunofluorescence staining was first performed to examine their expression in wild-type mice at postnatal month 1 (1M). In both the coronal and sagittal sections, NR2F1 exhibited a septal/dorsal high-temporal/ventral low expression pattern along the hippocampus (*Figure 1Aa, d, c, f, g, j, m, i, l, and o*), and its expression is highest in the dorsal CA1 region (*Figure 1Aa, d, c, f, j, and l*), where place cells are mainly located (*O'Keefe and Conway, 1978*; *O'Keefe and Dostrovsky, 1971*), and the dorsal DG, where there are adult neural stem cells (NSCs) (*Gould and Cameron, 1996*). However, the expression of NR2F2 was high in the temporal/ventral hippocampus but was barely detected in the septal/dorsal part of the hippocampus (*Figure 1Ab, c, e, f, h, i, k, l, n, and o*). The dorsal-high NR2F1 and ventral-high NR2F2 expression profiles were further verified in the postnatal hippocampi at 1M by western blotting assays (*Figure 1Ap and q*). At embryonic day 10.5 (E10.5), NR2F1 was detected in the dorsal pallium (DP) laterally and NR2F2 was expressed in the MP and CH medially (*Figure 1—figure supplement 1Aa and b*). At E11.5 and E12.5, the expression of NR2F2 remained in the CH (*Figure 1—figure supplement 1Ac, d and Bb, c*). Interestingly, NR2F1 and NR2F2 generated complementary expression patterns in the hippocampal primordium with NR2F1 in the dorsal MP and NR2F2 in the ventral CH at E14.5 (*Figure 1—figure supplement 1Ba–f*). Additionally, septal/dorsal-high NR2F1 and temporal/ventral-high NR2F2 expression patterns were observed at postnatal day 0 (P0) (*Figure 1—figure supplement 1Bg–l*). The data above revealed that the differential expression patterns of *Nr2f1* and *Nr2f2* genes were generated and maintained along the dorsoventral axis in the early hippocampal primordium, the developing and postnatal hippocampus, indicating that they could play distinct roles in the mediation of the morphogenesis and functions of the hippocampus.

Next, to investigate the roles of *Nr2f* genes in the hippocampus, an RX^Cre mouse was used to excise the expression of the *Nr2f1* and/or *Nr2f2* genes (*Swindell et al., 2006*; *Tang et al., 2012*). The deletion efficiency of RXCre recombinase was verified by immunofluorescence assays. Compared with control mice, either *Nr2f2* or *Nr2f1* could be excised in the postnatal hippocampus of corresponding single-gene mutants at 1M (*Figure 1—figure supplement 1Ca–i*). In addition, compared with control mice, both the *Nr2f1* and *Nr2f2* genes were almost completely deleted in the hippocampal primordium in mutant mice at E14.5 (*Figure 1—figure supplement 1Cj–o*). Since the LacZ expression serves as an indicator for the deletion of *Nr2f2* (*Swindell et al., 2006*; *Tang et al., 2012*), we performed immunofluorescence staining with antibodies against NR2F2 and LacZ on the sagittal sections of RX^Cre/+; *Nr2f2*^F/+ and RX^Cre/+; *Nr2f2*^F/F mice at E11.5. NR2F2 was readily detected at the hippocampal primordium of the heterozygous mutant embryo at E11.5 (*Figure 1—figure supplement 1Da, c, and g*); in contrast, the expression of *Nr2f2* was significantly reduced in the homozygous mutant (*Figure 1—figure supplement 1Dd, f, and j*). In addition, compared with the heterozygous mutant embryo (*Figure 1—figure supplement 1Db, c, and h*), the LacZ signals clearly increased in the hippocampal primordium of the homozygous mutant embryo at E11.5 (*Figure 1—figure supplement 1De, f, and k*), suggesting that RX-Cre recombinase can efficiently excise the *Nr2f2* gene in the hippocampal primordium as early as E11.5. Intriguingly, we observed that the expression of *Nr2f1* increased in the caudal hippocampal primordium of the *Nr2f2* homozygous mutant embryo at E11.5 (*Figure 1—figure supplement 1Di and l*), indicating that similar to the observations in the early optic cup (*Tang et al., 2012*), *Nr2f1* and *Nr2f2* genes could be partially compensate with each other in the developing hippocampal primordium. All the data above show that RXCre recombinase could efficiently excise *Nr2f1* and/or *Nr2f2* in the early developing and postnatal hippocampus.

### The *Nr2f2* gene is required for the appropriate morphogenesis of the ventral hippocampus but not of the dorsal hippocampus

Given that the *Nr2f2* gene is highly and specifically expressed in the postnatal ventral hippocampus and the CH of the hippocampal primordium (*Figure 1* and *Figure 1—figure supplement 1*), we asked

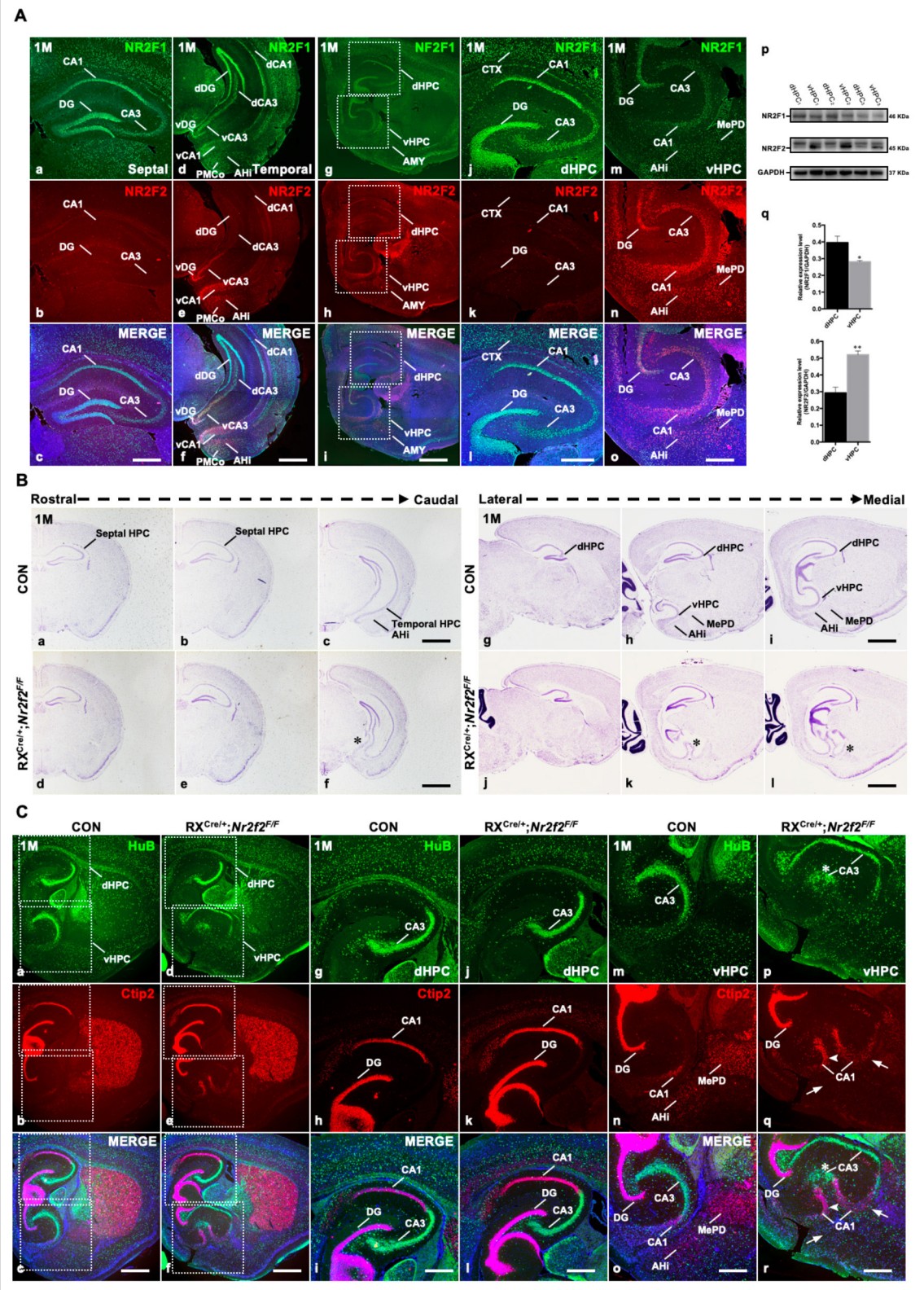

**Figure 1.** Duplicated CA1 and CA3 domains are generated in the ventral hippocampus of RX[Cre/+]; *Nr2f2*[F/F] mutant mice. (**A**) The expression of NR2F1 (**a, d, g, j, m**) and NR2F2 (**b, e, h, k, n**) in coronal sections (**a–f**) and sagittal sections (**g–o**) of the hippocampus at postnatal month 1 (1M); representative western blots and quantitative densitometry data for the expression of NR2F1 and NR2F2 in the dorsal and ventral hippocampus at 1M (**p–q**). Data are presented as mean ± SEM. *Student's t test* was used in q, *P<0.05, **P<0.01; n represents separate experiments, n=3. (**B**) In coronal sections along the

*Figure 1 continued on next page*

*Figure 1 continued*

rostrocaudal axis (**a–f**) and sagittal sections along the lateral-medial axis (**g–l**) of the hippocampus in mutant mice, compared with that in control mice (**a–c, g–i**), the ectopic CA-like structure, indicated by the star, was observed in the ventral region in *Nr2f2* gene mutant (RX$^{Cre/+}$; *Nr2f2$^{F/F}$*) mice at 1M (**d–f, j–l**). (**C**) The expression of HuB and Ctip2 in the corresponding inserted area in (**a–f**) under a high-magnification objective lens at 1M (**g–r**); compared with those of control mice (**a–c, g–i, m–o**), the duplicated HuB-positive CA3 domain, indicated by the star, and Ctip2-positive domains, indicated by the arrowhead, were specifically observed in the ventral hippocampus (**d–f, p–r**) but not in the dorsal hippocampus (**d–f, j–l**) of *Nr2f2* mutant mice at 1M; Ctip2-positive AHi and MePD amygdaloid nuclei were barely observed in the *Nr2f2* mutant mice, indicated by the arrows, instead of the ectopic CA domains at the prospective amygdaloid regions (**e–f, q–r**). AHi, amygdalohippocampal area; AMY, amygdala nuclei; CTX, cortex; dCA1, dorsal CA1; dCA3, dorsal CA3; dDG, dorsal dentate gyrus; dHPC, dorsal hippocampus; MePD, posterodorsal part of the medial amygdaloid nucleus; PMCo, posteromedial cortical amygdaloid nucleus; vCA1, ventral CA1; vCA3, ventral CA3; vDG, ventral dentate gyrus; vHPC, ventral hippocampus. Scale bars, (**Aa–c, Ad–f, Aj–o, Cg–r**), 100 μm; (**Ag–i, Ba–l, Ca–f**), 200 μm.

The online version of this article includes the following source data and figure supplement(s) for figure 1:

**Source data 1.** The Nissl staining results of the control and RX$^{Cre/+}$; *Nr2f2$^{F/F}$* mutant mice at 1M (part 1); the expression of HuB and Ctip2 in the hippocampus of the control and RX$^{Cre/+}$; *Nr2f2$^{F/F}$* mutant mice at 1M (part 1).

**Source data 2.** The expression of NR2F1 and NR2F2 in coronal sections and sagittal sections of the mouse brain at 1M (part 1); the expression of HuB and Ctip2 in the hippocampus of the control and RX$^{Cre/+}$; *Nr2f2$^{F/F}$* mutant mice at 1M (part 2).

**Source data 3.** The expression of NR2F1 and NR2F2 in coronal sections and sagittal sections of the mouse brain at 1M (part 2).

**Source data 4.** The expression of NR2F1 and NR2F2 in sagittal sections of the mouse brain at 1M (part 1).

**Source data 5.** The expression of NR2F1 and NR2F2 in sagittal sections of the mouse brain at 1M (part 2); the Nissl staining results of the control and RX$^{Cre/+}$; *Nr2f2$^{F/F}$* mutant mice at 1M (part 2).

**Source data 6.** The expression of NR2F1 and NR2F2 in sagittal sections of the mouse brain at 1M (part 3); western blots data for the expression of NR2F1 and NR2F2 in the dorsal and ventral hippocampus at 1M; the Nissl staining results of the control and RX$^{Cre/+}$; *Nr2f2$^{F/F}$* mutant mice at 1M (part 3).

**Figure supplement 1.** The expression of *Nr2f* genes in the early developing hippocampus and different conditional knock mouse models.

**Figure supplement 1—source data 1.** The expression of *Nr2f1* and *Nr2f2* genes in the developing hippocampus at E12.5; the deletion efficiency of RXCre recombinase in the hippocampus of RX$^{Cre/+}$; *Nr2f2$^{F/F}$*, RX$^{Cre/+}$; *Nr2f1$^{F/F}$*, and RX$^{Cre/+}$; *Nr2f1$^{F/F}$*; *Nr2f2$^{F/F}$* mice (part 1).

**Figure supplement 1—source data 2.** The expression of *Nr2f1* and *Nr2f2* genes in the telencephalon at E10.5, E11.5, E14.5, and P0; the deletion efficiency of RXCre recombinase in the hippocampus of RX$^{Cre/+}$; *Nr2f2$^{F/F}$*, RX$^{Cre/+}$; *Nr2f1$^{F/F}$*, and RX$^{Cre/+}$; *Nr2f1$^{F/F}$*; *Nr2f2$^{F/F}$* mice (part 2).

whether the *Nr2f2* gene is required for the appropriate morphogenesis of the hippocampus, particularly the ventral hippocampus. To answer this question, we conducted Nissl staining with samples from the *Nr2f2* single-gene (RX$^{Cre/+}$; *Nr2f2$^{F/F}$*) knockout mouse model. In coronal sections, compared with the control at 1M, the septal hippocampus was normal; unexpectedly, an ectopic CA-like region was observed medially in the temporal hippocampus in the *Nr2f2* mutant, where the prospective posterior part of the medial amygdaloid (MeP) nucleus was situated, indicated by the star (*Figure 1Ba–f*). The presence of the ectopic CA-like region in the ventral but not dorsal hippocampus of the mutant was further confirmed by the presence of the prospective MeP and amygdalohippocampal area (AHi) in sagittal sections, as indicated by the star (*Figure 1Bg–l*). Furthermore, immunofluorescence assays were performed to verify whether specific lineages of the hippocampus were altered with sagittal sections, in which the subregions of both the dorsal and ventral hippocampus were well displayed and distinguished. Ctip2 is a marker for CA1 pyramidal neurons and DG granule neurons, and HuB is a marker for CA3 pyramidal cells (*Sugiyama et al., 2014*). The dorsal hippocampus appeared normal in both the control and mutant mice at 1M (*Figure 1Ca–l*). Nonetheless, compared with the observations in control mice, an ectopic HuB-positive CA3 pyramidal neuron lineage, indicated by the star, and a duplicated Ctip2-positive CA1 pyramidal neuron lineage, indicated by the arrowhead, were observed in the ventral hippocampal area in the mutant (*Figure 1Ca–f and m–r*), revealing that there were ectopic CA1 and CA3 lineages in the *Nr2f2* mutants.

Consistent with the previous report (*Leid et al., 2004*), the expression of Ctip2 was detected in the amygdala including the AHi and posteromedial cortical amygdaloid nucleus (PMCo); in addition, Ctip2 was also highly expressed in the dorsal part of the MeP (MePD) in the control (*Figure 1Cb and n*).

Intriguingly, compared with the controls at 1M, there were ectopic CA domains in the mutant ventral hippocampus with the expense of the Ctip2-positive AHi and MePD amygdaloid nuclei (*Figure 1Ce and q*), indicated by the arrows. Clearly, all the data above suggested that the *Nr2f2* gene is necessary to ensure the appropriate morphogenesis of the ventral hippocampus.

At early embryonic stages, *Nr2f2* was preferentially expressed in the CH (*Figure 1—figure supplement 1Ab, d and Bb, e*), the organizer of the hippocampus, and at postnatal 1-month-old (1M) stage, *Nr2f2* was also highly expressed in some amygdala nuclei such as the AHi and medial amygdaloid nucleus, which are adjacent to the ventral/temporal hippocampus (*Figure 1Ae, h, and n*; *Tang et al., 2012*). We would like to investigate the correlation of the CH and/or amygdala anlage with the duplicated ventral hippocampal domains in the *Nr2f2* mutant in detail in our future study. The observations above suggest that *Nr2f2* is not only specifically expressed in the ventral hippocampus but is also required for morphogenesis and probably the function of the ventral hippocampus. Since the ventral hippocampus participates in the regulation of emotion and stress, mutations in the *Nr2f2* gene lead to CHDs, and the formation of the ventral hippocampus is disrupted in *Nr2f2* mutant mice at 1M, we wondered whether CHD patients with *Nr2f2* mutations also exhibit symptoms associated with psychiatric disorders such as depression, anxiety, or schizophrenia.

## The *Nr2f1* gene is required for the specification and differentiation of the dorsal CA1 identity

Next, we asked whether the deletion of the *Nr2f1* gene by RX-Cre also affected the development of the hippocampus. Consistent with the previous finding in *Emx1^{Cre/+}*; *Nr2f1^{F/F}* mutant mice (*Flore et al., 2017*), it was the septal/dorsal hippocampus, not the temporal/ventral hippocampus, that was specifically shrunken in both coronal (*Figure 2Aa–f*) and sagittal sections (*Figure 2Ag–i*) of RX^{Cre/+}; *Nr2f1^{F/F}* mutant mice. Then, we asked whether the loss of the *Nr2f1* gene caused abnormal specification and differentiation of hippocampal lineages. Compared with the control mice, *Nr2f1* mutant mice had fewer HuB-positive CA3 pyramidal neurons, as indicated by the star; intriguingly, Ctip2-positive CA1 pyramidal neurons failed to be detected, as indicated by the arrowhead, with Ctip2-positive DG granule neurons unaltered in the dorsal hippocampus (*Figure 2Ba–l*). The loss of the dorsal CA1 pyramidal neuron identity in mutant mice was further confirmed by Wfs1, another dorsal CA1 pyramidal neuron-specific marker (*Takeda et al., 2001*; *Figure 2Ca–r*). Nonetheless, the HuB-positive and Ctip2-positive lineages were comparable in the ventral hippocampus between the control and mutant mice (*Figure 2Ba–f and m–r*), even though the low expression of NR2F1 was detected there. Indeed, *Nr2f1* is not only expressed at the highest level in the dorsal CA1 but is also required for the specification and differentiation of dorsal CA1 pyramidal neurons, among which place cells are essential for learning and memory (*O'Keefe and Conway, 1978*; *O'Keefe and Dostrovsky, 1971*).

Given that dysplasia of the dorsal hippocampus was generated in both the *Emx1^{Cre}* and RX^{Cre} models (*Flore et al., 2017*; *Figure 2Aa–l*), we asked whether the development of dorsal CA1 pyramidal neurons was also abolished in *Emx1^{Cre/+}*; *Nr2f1^{F/F}* mutant mice. To answer this question, immunofluorescence staining was conducted first. Compared with that in the control mice, the proportions of either the Wfs1- or Ctip2-positive CA1 domain were reduced in the mutant mice at 3M (*Figure 2—figure supplement 1Aa–h*), indicating that the differentiation of the dorsal CA1 pyramidal neurons was also compromised in the *Emx1^{Cre}* model, although it was less severe than that in the RX^{Cre} model. To make our findings more consistent with previous studies, we further conducted experiments with the *Emx1^{Cre}* model. Afterward, to investigate the fine structure of the dorsal CA1 pyramidal neurons, Golgi staining was performed. Compared with those of control mice, the numbers of secondary dendrites and branch points of both the apical and basal dendrites were significantly reduced in the dorsal CA1 pyramidal neurons of mutant mice at 3M (*Figure 2—figure supplement 1Ba–e*). Then, the dorsal hippocampus-related spatial learning and memory behavior test, the Morris water maze, was performed (*Vorhees and Williams, 2006*). Consistent with a previous report (*Flore et al., 2017*), spatial learning and memory function were significantly impaired in adult *Emx1^{Cre/+}*; *Nr2f1^{F/F}* mice compared with the control mice (*Figure 2—figure supplement 1C*). The data above suggest that *Nr2f1* is vital for the morphogenesis, lineage specification, and spatial learning and memory of the dorsal hippocampus, and particularly, the compromised dorsal CA1 lineage could contribute to the phenotypes associated with neurodevelopmental disorders, including ID or ASD.

## The *Nr2f1* and *Nr2f2* genes coordinate to ensure the genesis of the hippocampus

Given that the loss of either *Nr2f1* or *Nr2f2* leads to dysplasia of the dorsal or ventral hippocampus, respectively, we asked whether these genes compensate for each other to regulate the morphogenesis

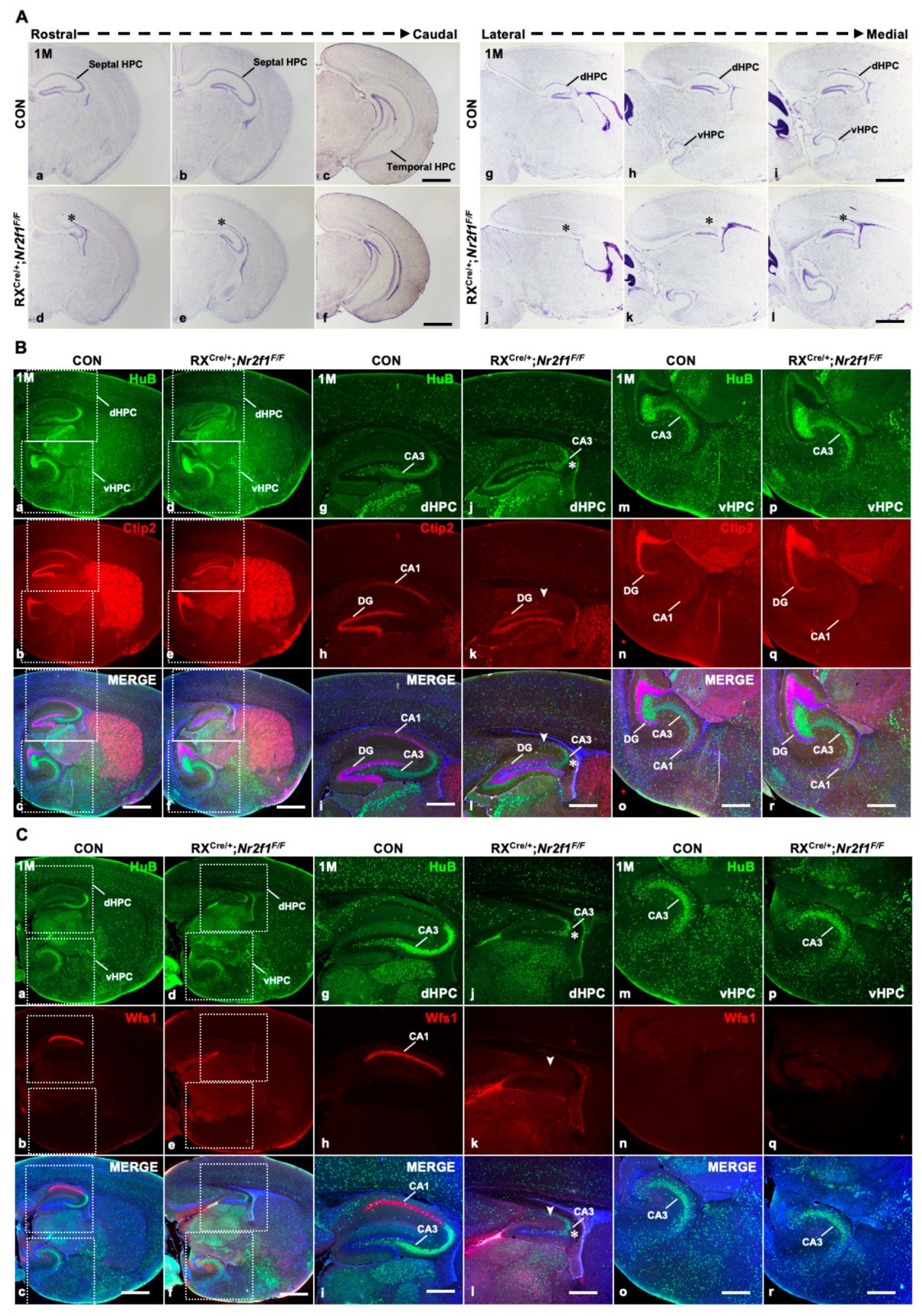

**Figure 2.** The specification and differentiation of the dorsal CA1 lineage failed with the dysplastic dorsal hippocampus in RX[Cre/+]; *Nr2f1[F/F]* mutant mice. (**A**) In coronal sections along the rostrocaudal axis (**a–f**) and sagittal sections along the lateral–medial axis (**g–l**) of the hippocampus, compared with that of control mice (**a–c, g–i**), the dorsal hippocampus was shrunken, indicated by the star, in *Nr2f1* gene mutant (RX[Cre/+]; *Nr2f1[F/F]*) mice at 1M (**d–f, j–l**). (**B**) The expression of HuB and Ctip2 in the corresponding inserted area in (**a–f**) under a high-magnification objective lens at 1M (**g–r**); compared

*Figure 2 continued on next page*

*Figure 2 continued*

with that of control mice (**a–c, g–i, m–o**), the HuB-positive CA3 domain was reduced in the dorsal hippocampus, especially the Ctip2-positive dorsal CA1, which was barely detected in *Nr2f1* mutant mice at 1M (**d–f, j–l**), while their expression in the ventral hippocampus was comparable between the controls and mutants (**d–f, p–r**). (**C**) The expression of HuB and Wfs1 in the corresponding inserted area in (**a–f**) under a high-magnification objective lens at 1M (**g–r**); the expression of HuB and the dCA1 marker Wfs1 in the control (**a–c, g–i, m–o**) and *Nr2f2* mutant mice (**d–f, j–l, p–r**) at 1M. Wfs1-positive dorsal CA1 could not be detected in *Nr2f1* mutant mice at 1M, as indicated by the arrowhead. dHPC, dorsal hippocampus; HPC, hippocampus; vHPC, ventral hippocampus. Scale bars, (**Aa–l, Ba–f, Ca–f**), 200 μm; (**Bg–r, Cg–r**), 100 μm.

The online version of this article includes the following source data and figure supplement(s) for figure 2:

**Source data 1.** The Nissl staining results of the control and RX^Cre/+^; *Nr2f1^F/F^* mutant mice at 1M (part 1).

**Source data 2.** The Nissl staining results of the control and RX^Cre/+^; *Nr2f1^F/F^* mutant mice at 1M (part 2); the expression of HuB and Wfs1 in the hippocampus of the control and RX^Cre/+^; *Nr2f1^F/F^* mutant mice at 1M.

**Source data 3.** The expression of HuB and Ctip2 in the hippocampus of the control and RX^Cre/+^; *Nr2f1^F/F^* mutant mice at 1M.

**Source data 4.** The expression of HuB, Wfs1, and Ctip2 in the hippocampus of the control and RX^Cre/+^; *Nr2f1^F/F^* mutant mice at 1M.

**Figure supplement 1.** Defects in *Emx1^Cre/+^*; *Nr2f1^F/F^* mutant mice.

**Figure supplement 1—source data 1.** The expression of Wfs1 and Ctip2 in the dorsal hippocampus of the control and *Emx1^Cre/+^*; *Nr2f1^F/F^* mutant mice at 3M; Golgi staining results of the dorsal hippocampal CA1 pyramidal neurons and Morris water maze behavior test data of the control and *Emx1^Cre/+^*; *Nr2f1^F/F^* mutant mice.

of the hippocampus. To answer this question, the RX^Cre/+^; *Nr2f1^F/F^*; *Nr2f2^F/F^* double-mutant mouse was generated, and a few homozygous double-gene mutant mice survived for approximately 3 weeks (3W). Nonetheless, the reason for the lethality of the double-mutant mice is still unknown. Nissl staining data showed that compared with that of control mice, the septal hippocampus was severely shrunken, as indicated by the star, and the temporal hippocampus was barely observed in the double-mutant mouse brains (*Figure 3Aa–h*). Unexpectedly, an ectopic nucleus was observed in the region of the prospective temporal hippocampus, indicated by the arrowhead, in the double-mutant mice (*Figure 3Ag–h*). In addition, compared with those of controls, the regions with HuB-positive CA3 pyramidal neurons and Ctip2-positive or Prox1-positive DG granule neurons were diminished in the double mutants; in particular, Ctip2-positive dorsal CA1 pyramidal neurons could not be detected in the double mutants (*Figure 3Ba–l*). Furthermore, compared with the domains of controls, no HuB-positive, Ctip2-positive, or Prox1-positive domains could be detected in the prospective temporal hippocampus in the double mutants (*Figure 3Ca–l*). The results above suggest that the *Nr2f1* and *Nr2f2* genes coordinate with each other to mediate the appropriate morphogenesis of the entire hippocampus.

### *Nr2f* genes and adult neurogenesis in the hippocampus

Given that the *Nr2f1* or *Nr2f2* gene was highly expressed in the dorsal or ventral DG, respectively (*Figure 1A*), and that RX-Cre recombinase could efficiently delete either gene in the DG (*Figure 1—figure supplement 1Ca–i*), we asked whether the loss of the *Nr2f1* or/and -*Nr2f2* gene in the DG would affect hippocampal adult neurogenesis. To answer this question, the ventral DG, dorsal DG, and septal DG were chosen to perform immunofluorescence assays in the *Nr2f2* mutant, *Nr2f1* mutant, and double-mutant models, respectively. Adult NSCs in the subgranular zone (SGZ) of the DG express both GFAP and Nestin, and newborn granule neurons express Dcx (*Gao et al., 2007*). The numbers of NSCs and newborn neurons in the SGZ of the DG were comparable between control mice and either the *Nr2f2* or *Nr2f1* mutant mice (*Figure 3—figure supplement 1Aa–p and Ba–h*); nonetheless, compared with those of control mice, the numbers of the NSCs and newborn granule neurons in the SGZ of the DG were reduced in the double mutants (*Figure 3—figure supplement 1Aq–x and Bi–l*), and the reduction in both lineages was significant (*Figure 3—figure supplement 1Ca and b*). The data above suggest that *Nr2f* genes may coordinate with each other to execute essential functions for appropriate hippocampal adult neurogenesis in the DG.

### Hippocampal trisynaptic connectivity was impaired in postnatal *Nr2f2* single-, *Nr2f1* single-, and double-mutant mice at about 1M

Given that dysplasia of the hippocampus was observed in all three mouse models, we asked whether the connectivity of the hippocampal trisynaptic circuit associated with the DG, CA3, and CA1 regions

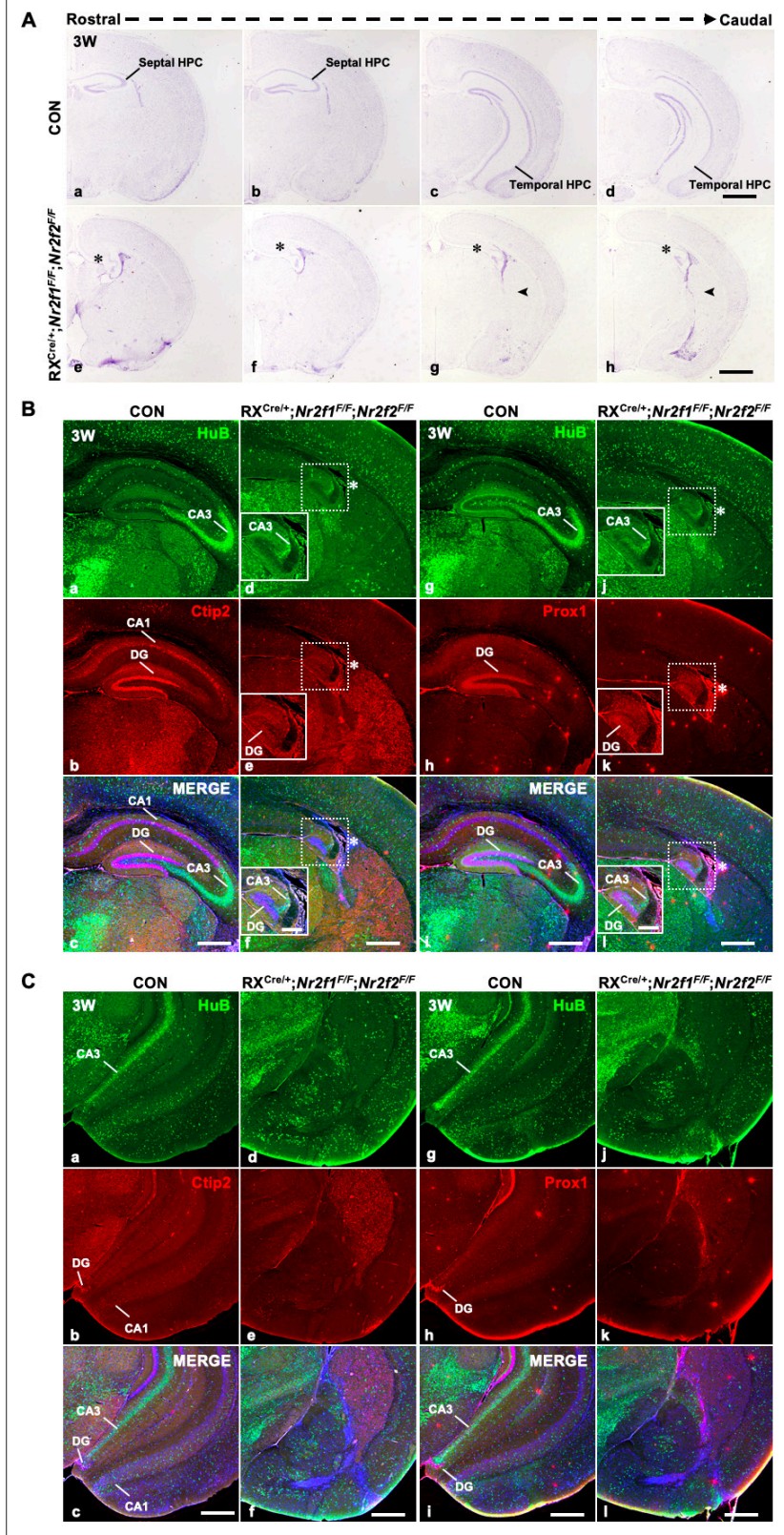

**Figure 3.** Defects in the hippocampus in RX^Cre/+; *Nr2f1*^F/F; *Nr2f2*^F/F double-gene mutant mice. (**A**) In coronal sections along the rostrocaudal axis, compared with control mice (**a–d**), the hippocampus was atrophic in RX^Cre/+; *Nr2f1*^F/F; *Nr2f2*^F/F double-mutant mice, indicated by the star, and an ectopic unknown nucleus was observed in the caudal plates, indicated by the arrowhead (**e–h**). (**B**) Compared with that of control mice (**a–c, g–i**), the expression of HuB,

*Figure 3 continued on next page*

*Figure 3 continued*

Ctip2, and Prox1 was decreased in the hippocampus of *Nr2f1/2* double-gene mutant mice at 3 weeks postnatal (3W) (**d–f, j–l**). (**C**) Compared with that of control mice (**a–c, g–i**), the expression of HuB could not be detected in the presumptive CA3 domain, and the expression of Ctip2 or Prox1 could not be detected in the presumptive DG domain of the prospective ventral hippocampus of RX^Cre/+^; *Nr2f1*^F/F^; *Nr2f2*^F/F^ double-mutant mice. Scale bars, (**Aa–h**), 200 μm; (**Ba–l, Ca–l**), 100 μm; (**Bd-f** (insets), **Bj-l** (insets)), 400 μm.

The online version of this article includes the following source data and figure supplement(s) for figure 3:

**Source data 1.** The Nissl staining results of the control and RX^Cre/+^; *Nr2f1*^F/F^; *Nr2f2*^F/F^ double-gene mutant mice at 3W.

**Source data 2.** The expression of HuB and Ctip2 in the hippocampus of the control and RX^Cre/+^; *Nr2f1*^F/F^; *Nr2f2*^F/F^ double-gene mutant mice at 3W.

**Source data 3.** The expression of HuB and Prox1 in the hippocampus of the control and RX^Cre/+^; *Nr2f1*^F/F^; *Nr2f2*^F/F^ double-gene mutant mice at 3W.

**Source data 4.** The expression of HuB, Ctip2, and Prox1 in the hippocampus of the control and RX^Cre/+^; *Nr2f1*^F/F^; *Nr2f2*^F/F^ double-gene mutant mice at 3W.

**Figure supplement 1.** Adult neurogenesis was abnormal in the hippocampi of *Nr2f1/2* double-gene mutant mice.

**Figure supplement 1—source data 1.** The expression of GFAP, Nestin, and Dcx in the subgranular zone (SGZ) of ventral dentate gyrus (vDG) in the control and RX^Cre/+^; *Nr2f2*^F/F^ mutant mice at 1M, in the SGZ of dorsal dentate gyrus (dDG) in the control and RX^Cre/+^; *Nr2f1*^F/F^ mutant mice at 1M, and in the SGZ of DG in the control and RX^Cre/+^; *Nr2f1*^F/F^; *Nr2f2*^F/F^ double-gene mutant mice at 3W (part 1).

**Figure supplement 1—source data 2.** The expression of GFAP, Nestin, and Dcx in the subgranular zone (SGZ) of ventral dentate gyrus (vDG) in the control and RX^Cre/+^; *Nr2f2*^F/F^ mutant mice at 1M, in the SGZ of dorsal dentate gyrus (dDG) in the control and RX^Cre/+^; *Nr2f1*^F/F^ mutant mice at 1M, and in the SGZ of DG in the control and RX^Cre/+^; *Nr2f1*^F/F^; *Nr2f2*^F/F^ double-gene mutant mice at 3W (part 2).

**Figure supplement 1—source data 3.** The expression of GFAP and Nestin in the subgranular zone (SGZ) of ventral dentate gyrus (vDG) in the control and RX^Cre/+^; *Nr2f2*^F/F^ mutant mice at 1M, in the SGZ of dorsal dentate gyrus (dDG) in the control and RX^Cre/+^; *Nr2f1*^F/F^ mutant mice at 1M, and in the SGZ of DG in the control and RX^Cre/+^; *Nr2f1*^F/F^; *Nr2f2*^F/F^ double-gene mutant mice at 3W; quantitative analysis of GFAP/Nestin-positive cells in the SGZ of DG in the control and RX^Cre/+^; *Nr2f1*^F/F^; *Nr2f2*^F/F^ double-gene mutant mice at 3W.

**Figure supplement 1—source data 4.** The expression of GFAP, Nestin, and Dcx in the subgranular zone (SGZ) of ventral dentate gyrus (vDG) in the control and RX^Cre/+^; *Nr2f2*^F/F^ mutant mice at 1M, in the SGZ of dorsal dentate gyrus (dDG) in the control and RX^Cre/+^; *Nr2f1*^F/F^ mutant mice at 1M, and in the SGZ of DG in the control and RX^Cre/+^; *Nr2f1*^F/F^; *Nr2f2*^F/F^ double-gene mutant mice at 3W (part 3); quantitative analysis of Dcx-positive cells in the SGZ of DG in the control and RX^Cre/+^; *Nr2f1*^F/F^; *Nr2f2*^F/F^ double-gene mutant mice at 3W.

---

(*Amaral, 1993*) was normal in these models. To answer this question, the components of the trisynaptic circuit were characterized in the ventral hippocampus of *Nr2f2* mutants, the dorsal hippocampus of *Nr2f1* mutants, and the septal hippocampus of double mutants. Calretinin is a marker of mossy cells, Calbindin is a marker of mossy fibers, and SMI312 is a marker of Schafer collaterals (*Flore et al., 2017*). Compared with those of controls (*Figure 4Aa, b, e, f, i, and j*), the numbers of Calretinin-positive mossy cells were reduced, Calbindin-positive mossy fibers were longer but thinner, and SMI312-positive Schafer collaterals were thinner and discontinued in the ventral hippocampus of *Nr2f2* mutants at 1M (*Figure 4Ac, d, g, h, k, and l*). In addition, similar to the previous report (*Flore et al., 2017*), the numbers of Calretinin-positive mossy cells were decreased, Calbindin-positive mossy fibers were shorter and thinner, and SMI312-positive Schafer collaterals were barely detected in the dorsal hippocampus of *Nr2f1* mutants (*Figure 4Ac, d, g, h, k, and l*), compared with those of controls at 1M (*Figure 4Ba, b, e, f, i, and j*). Moreover, compared with those of control mice (*Figure 4Ca, b, e, f, i, and j*), the numbers of Calretinin-positive mossy cells were reduced; both Calbindin-positive mossy fibers and SMI312-positive Schafer collaterals were barely detected in the prospective septal hippocampus of the double mutants at 3W (*Figure 4Cc, d, g, h, k, and l*). Clearly, without both *Nr2f* genes, the connectivity of the hippocampal trisynaptic circuit was abolished more severely. The observations above revealed that the formation of the trisynaptic circuit, which is one of the fundamental characteristics of hippocampal neurophysiology (*Basu and Siegelbaum, 2015*), was abnormal in all three mouse models, indicating that both the morphology and functions of the hippocampus are most likely compromised in the loss of the *Nr2f1* and/or *Nr2f2* gene.

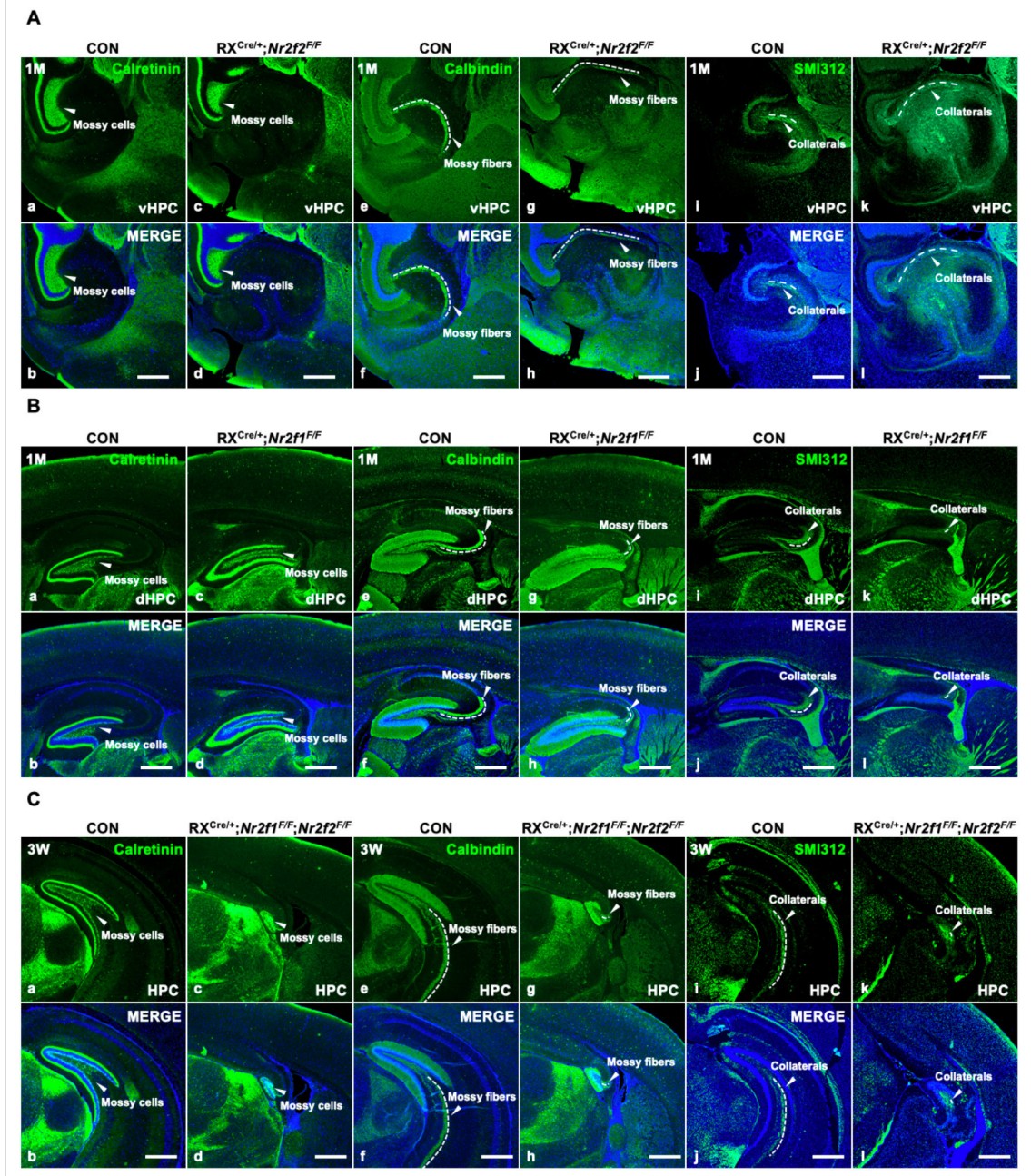

**Figure 4.** The impairment of hippocampal trisynaptic connectivity in *Nr2f2* single-gene, *Nr2f1* single-gene, and *Nr2f1/2* double-gene mutant mice. (**A**) The expression of Calretinin, Calbindin, and SMI312 in the ventral hippocampus of the control (**a, b, e, f, i, j**) and *Nr2f2* single-gene mutant mice (**c, d, g, h, k, l**). (**B**) The expression of Calretinin, Calbindin, and SMI312 in the dorsal hippocampus of the control (**a, b, e, f, i, j**) and *Nr2f1* single-gene mutant mice (**c, d, g, h, k, l**). (**C**) The expression of Calretinin, Calbindin, and SMI312 in the hippocampus of the control (**a, b, e, f, i, j**) and *Nr2f1/2* double-gene mutant mice (**c, d, g, h, k, l**). dHPC, dorsal hippocampus; HPC, hippocampus; vHPC, ventral hippocampus. Scale bars, (**Aa–l, Ba–l, Ca–l**), 100 μm.

The online version of this article includes the following source data for figure 4:

**Source data 1.** The expression of Calretinin, Calbindin, and SMI312 in the ventral hippocampus of the control and RX^Cre/+; *Nr2f2^F/F* single-gene mutant mice at 1M.

**Source data 2.** The expression SMI312 in the dorsal hippocampus of the control and RX^Cre/+; *Nr2f1^F/F* single-gene mutant mice at 1M, and the expression of Calretinin and Calbindin in the hippocampus of the control and RX^Cre/+; *Nr2f1^F/F*; *Nr2f2^F/F* double-gene mutant mice at 3W.

**Source data 3.** The expression of Calretinin and Calbindin in the dorsal hippocampus of the control and RX^Cre/+; *Nr2f1^F/F* single-gene mutant mice at 1M, and the expression of SMI312 in the hippocampus of the control and RX^Cre/+; *Nr2f1^F/F*; *Nr2f2^F/F* double-gene mutant mice at 3W.

## The expression of several essential regulatory genes associated with early hippocampal development was abnormal in double mutants

Given that the hippocampus was almost completely diminished in double mutants, we asked how *Nr2f* genes participated in the regulation of the early morphogenesis of the hippocampus. To answer this question, total RNA isolated from the whole telencephalons of control (n = 5) and double-mutant (n = 3) embryos at E11.5 was used to generate cDNA, and then real-time quantitative PCR (RT-qPCR) assays were performed. As expected, compared with that of control mice, the expression of *Nr2f1* and *Nr2f2* was reduced significantly in the double-mutant mice (*Figure 5A*). Then, we mainly focused on the intrinsic regulatory networks by analyzing the expression profiles of two groups of transcription factor genes. The *Foxg1*, *Gli3*, *Lhx2*, *Otx1*, *Otx2*, and *Pax6* genes, which are highly related to the early patterning of the dorsal telencephalon (*Hébert and Fishell, 2008*), were in the first group; *Axin2*, *Emx1*, *Emx2*, *Lef1*, *Lhx5*, and *Tcf4* genes, which are associated with early hippocampal development (*Galceran et al., 2000*; *Moore and Iulianella, 2021*; *Tole et al., 2000*; *Yoshida et al., 1997*; *Zhao et al., 1999*), were in the other group. The expression of the *Foxg1*, *Gli3*, *Lhx2*, *Otx1*, *Otx2*, and *Pax6* genes was comparable between the controls and double mutants (*Figure 5A*), indicating that the early patterning of the dorsal telencephalon is largely unaltered. Compared with that of control mice, the expression of the *Axin2*, *Emx2*, *Lef1*, and *Tcf4* genes was normal in the double mutants; interestingly, the expression of the *Emx1* and *Lhx5* transcripts was decreased significantly in the double mutants at E11.5 compared to that in control mice (*Figure 5A*). Consistent with the downregulated expression of *Lhx5* transcripts in the double mutant, the expression of the Lhx5 protein was reduced in the CH in the double mutants at E11.5; moreover, the number of Lhx5-positive Cajal–Retzius cells decreased in the double-mutant embryos at E11.5, E13.5, and E14.5 (*Figure 5Ba–d, a'–d', a"–d", i–l, i'–l', q–t, and q'–t'*). The expression of *Lhx2* was expanded ventrally into the choroid plexus in the *Lhx5* null mutant mice (*Zhao et al., 1999*), indicating that *Lhx5* could inhibit *Lhx2* expression locally. Consistent with RT-qPCR data, the expression of Lhx2 was comparable between the control and double-mutant mice at E11.5 (*Figure 5Be–h and e'–h'*). Interestingly, the expression of the Lhx2 protein was increased in the hippocampal primordium in the *Nr2f* double-mutant mice at E13.5 and E14.5 (*Figure 5Bm–p, m'–p', u–x, and u'–x'*). The upregulation of Lhx2 expression is most likely associated with the reduced expression of the *Lhx5* gene.

Next, we asked whether neural precursor cells (NPCs), intermediate progenitor cells (IPCs), or newborn neurons were affected in the early development of the hippocampus in double-mutant mice. Sox2 is a marker for NPCs, Tbr2 is a marker for IPCs, and NeuroD1 is a marker for newborn neurons (*Yu et al., 2014*). The expression of Sox2 in the hippocampal regions was comparable between the control and double-mutant mice at E14.5 (*Figure 5Ca–d and a'–d'*), indicating that the generation of NPCs was normal. Nevertheless, compared with the control embryos, the numbers of Tbr2-positive IPCs and NeuroD1-positive newborn neurons were reduced in the double-mutant embryos (*Figure 5Ce–l and e'–l'*), and the reduction was significant (*Figure 5D*). Our observations were consistent with previous findings in *Lhx5* null mutant mice that the specification of the hippocampal NPCs was normal, but the later differentiation event was abolished (*Zhao et al., 1999*). All the data above suggest that *Nr2f* genes may cooperate to ensure the early morphogenesis of the hippocampus by regulating the appropriate expression levels of *Lhx5* and *Lhx2* genes. Nevertheless, we could not exclude other possibilities that *Nr2f* genes could also participate in the modulation of hippocampal development through *Emx1* or other genes.

## Discussion

In this study, we observed dorsal-high NR2F1 and ventral-high NR2F2 expression profiles in the postnatal hippocampus. The deletion of the *Nr2f2* gene led to duplicated CA1 and CA3 domains of the ventral hippocampus. The loss of *Nr2f1* resulted in the failed specification and differentiation of the dorsal CA1 pyramidal neuron lineage with a diminished dorsal hippocampus. Furthermore, the deficiency of both *Nr2f* genes caused atrophy of almost the entire hippocampus, accompanied by compromised generation of the CA1, CA3, and DG identities. In addition, the dorsal trisynaptic components, ventral trisynaptic components, or entire trisynaptic components were abolished in the corresponding *Nr2f1* gene mutant, *Nr2f2* gene mutant, or *Nr2f1/2* double-gene mutant mice.

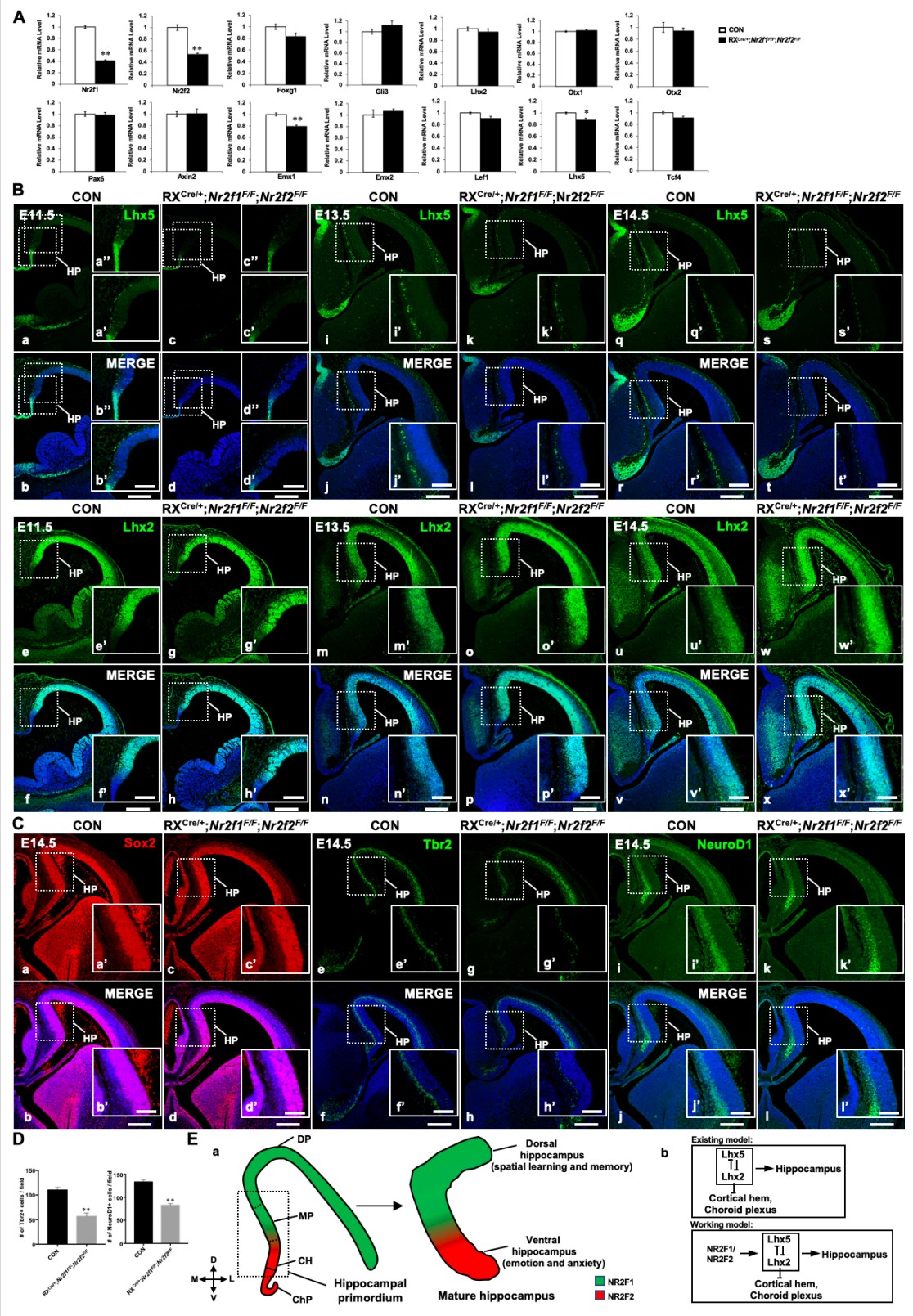

**Figure 5.** *Nr2f* genes regulate the expression of key genes associated with early hippocampal development. (**A**) The expression profiles of genes involved in hippocampal development in control and the double-mutant mice at E11.5. Data are presented as mean ± SEM. *Student's t test* was used in A, *P<0.05, **P<0.01; n represents separate experiments, n=3. (**B**) Compared with that of control mice (**a, b, a', b', a'', b'', i, j, i', j', q, r, q', r'**), the expression of Lhx5 was reduced in double-mutant mice at E11.5 (**c, d, c', d', c'', d''**), E13.5 (**k, l, k', l'**), and E14.5 (**s, t, s', t'**); the expression of Lhx2 was

*Figure 5 continued on next page*

*Figure 5 continued*

comparable between the control and double-mutant mice at E11.5 (**e–h, e'–h'**); and compared with that of control mice (**m, n, m', n', u, v, u', v'**), the expression of Lhx2 was increased in double-mutant mice at E13.5 (**o, p, o', p'**) and E14.5 (**w, x, w', x'**). (**C**) Compared with that of control mice (**a, b, a', b'**), the expression of Sox2 was normal in double-mutant mice at E14.5 (**c, d, c', d'**); compared with that of control mice (**e, f, e', f'**), the expression of Tbr2 was decreased in *Nr2f* mutant mice at E14.5 (**g, h, g', h'**); compared with that of control mice (**i, j, i', j'**), the expression of NeuroD1 was reduced in double-mutant mice at E14.5 (**k, l, k', l'**). (**D**) Quantitative analysis of Tbr2-positive cells and NeuroD1-positive cells in (**Ce'–h'**) and (**Ci'–l'**). Data are presented as mean ± SEM. *Student's t test* was used in D, **P<0.01; n represents separate experiments, n=3. (**E**), In the hippocampal primordium of the early embryo, *Nr2f1* is expressed dorsally in the MP, and *Nr2f2* is expressed ventrally in the CH. In the mature hippocampus, the expression of *Nr2f1* is higher in the dorsal hippocampus, which is related to spatial learning and memory, and the expression of *Nr2f2* is mainly in the ventral hippocampus, which is associated with emotion and anxiety (**a**). Our findings support a novel molecular mechanism by which *Nr2f1* and *Nr2f2* may cooperate to ensure the appropriate morphogenesis and functions of the hippocampus by modulating the *Lhx5-Lhx2* axis (**b**). CH, cortical hem; ChP, choroid plexus; DP, dorsal pallium; HP, hippocampal primordium; MP, medial pallium. Scale bars, (**Ba–x, Ca–l**), 200 μm; (**Ba'-x', Ba"-d", Ca'-l'**), 100 μm.

The online version of this article includes the following source data and figure supplement(s) for figure 5:

**Source data 1.** The expression of Lhx5 and Lhx2 in the telencephalon of the control and RX^Cre/+^; *Nr2f1^F/F^*; *Nr2f2^F/F^* double-mutant mice at E14.5; the expression of Tbr2 and NeuroD1 in the telencephalon of the control and RX^Cre/+^; *Nr2f1^F/F^*; *Nr2f2^F/F^* double-mutant mice at E14.5 (part 1).

**Source data 2.** The expression of Sox2, Tbr2, and NeuroD1 in the telencephalon of the control and RX^Cre/+^; *Nr2f1^F/F^*; *Nr2f2^F/F^* double-mutant mice at E14.5.

**Source data 3.** The expression profiles of genes involved in the hippocampal development of the control and double-mutant mice at E11.5; the expression of Lhx5 and Lhx2 in the telencephalon of the control and RX^Cre/+^; *Nr2f1^F/F^*; *Nr2f2^F/F^* double-mutant mice at E11.5 and E13.5; the expression of Tbr2 and NeuroD1 in the telencephalon of the control and RX^Cre/+^; *Nr2f1^F/F^*; *Nr2f2^F/F^* double-mutant mice at E14.5 (part 2); quantitative analysis of Tbr2-positive and NeuroD1-positive cells in the hippocampal primordium of the control and RX^Cre/+^; *Nr2f1^F/F^*; *Nr2f2^F/F^* double-mutant mice at E14.5.

**Figure supplement 1.** *Nr2f1* and *Nr2f2* genes coordinate to control distinct characteristics of the hippocampus.

**Figure supplement 1—source data 1.** Roles of *Nr2f1* and *Nr2f2* genes in the development and function of the hippocampus and the association with neurological disorders.

---

Moreover, *Nr2f* genes may cooperate to ensure the appropriate morphogenesis and function of the hippocampus by regulating the *Lhx5-Lhx2* axis.

## *Nr2f2* governs the distinct characteristics of the ventral hippocampus

Sixty years ago, the pioneering work of Milner and her colleagues discovered the essential role of the hippocampus in declarative memory (*Penfield and Milner, 1958*; *Scoville and Milner, 1957*). Recently, accumulating evidence has supported the Moser theory that the hippocampus is a heterogeneous structure with distinct characteristics of gene expression, connectivity, and function along its dorsoventral axis (*Bast, 2007*; *Fanselow and Dong, 2010*; *Moser and Moser, 1998*; *Strange et al., 2014*). The dorsal hippocampus marked in blue, in which gene expression is similar to the neocortex, serves the 'cold' cognitive function associated with declarative memory and spatial navigation, and the ventral hippocampus marked in red, in which gene expression is close to the hypothalamus and amygdala, corresponds to the 'hot' affective states related to emotion and anxiety (*Figure 5—figure supplement 1*). The ventral hippocampus generates direct connectivity with the amygdala, hypothalamus, medial prefrontal cortex (mPFC), and olfactory bulb (*Cenquizca and Swanson, 2007*; *Hoover and Vertes, 2007*; *Kishi et al., 2000*; *Pitkänen et al., 2000*; *Roberts et al., 2007*). Nonetheless, thus far, the molecular and cellular mechanism of how the morphogenesis, connectivity, and function of the ventral hippocampus is achieved has been largely unclear.

*Nr2f2*, a nuclear receptor gene associated with heart disease (*Al Turki et al., 2014*; *High et al., 2016*), was highly and exclusively expressed in the ventral hippocampus in 1-month-old mice and was expressed ventrally in the CH of the hippocampal primordium in mouse embryos (*Figure 1*, *Figure 1—figure supplement 1*, and *Figure 5Ea*), indicating that the *Nr2f2* gene may participate in the regulation of the development and function of the ventral hippocampus. First, deficiency of the *Nr2f2* gene led to the duplication of the CA1 and CA3 domains of the ventral hippocampus but not the dorsal hippocampus, which was confirmed both morphologically and molecularly (*Figure 1*). Second, the formation of the trisynaptic circuit was specifically abolished in the ventral hippocampus of *Nr2f2* mutants (*Figure 4*), indicating that the intrahippocampal circuit, information transfer, and function of the ventral hippocampus could be compromised. Third, the ventral hippocampus generates neural circuits with the mPFC, amygdala, nucleus accumbens, and hypothalamus, which are associated with anxiety/behavioral inhibition, fear processing, pleasure/reward seeking, and the neuroendocrine

system, respectively (*Anacker and Hen, 2017*; *Baik, 2020*; *Bryant and Barker, 2020*; *Cenquizca and Swanson, 2007*; *Herman et al., 2016*; *Kishi et al., 2000*; *O'Leary and Cryan, 2014*; *Pitkänen et al., 2000*). These ventral hippocampal projections may be important for processing information related to emotion and anxiety. Intriguingly, our previous studies revealed that *Nr2f2* is required for the development of the hypothalamus, amygdala, and olfactory bulb (*Feng et al., 2017*; *Tang et al., 2012*; *Zhou et al., 2015*), all of which generate functional neural circuits with the ventral hippocampus (*Fanselow and Dong, 2010*). Particularly, both the hypothalamus and amygdala are also diminished in RX^Cre/+; *Nr2f2*^F/F mutant mice (*Feng et al., 2017*; *Tang et al., 2012*), indicating that their interconnectivities with the ventral hippocampus are abnormal. Thus, all the findings above suggest that *Nr2f2* is a novel and essential intrinsic regulator that controls the morphogenesis, connectivity, and function of the ventral hippocampus.

Given that mutations of *Nr2f2* are highly associated with CHDs (*Al Turki et al., 2014*), the expression of *Nr2f2* is also confined to the ventral hippocampus in human embryos (*Alzu'bi et al., 2017*), and *Nr2f2* gene is required for the distinct characteristics of the ventral hippocampus in mouse (*Figures 1 and 4*), we wondered whether CHD patients carrying mutations of *Nr2f2* also display symptoms of psychiatric disorders, such as depression, anxiety, or schizophrenia, related to the ventral hippocampus. In our future study, we would like to generate hippocampus-specific or hippocampal subdomain-specific conditional knockout models to dissect distinct roles of the *Nr2f2* gene in the hippocampus, particularly in the ventral hippocampus, in detail.

## The *Nr2f1* gene is required for the specification and differentiation of dorsal CA1 pyramidal neurons

The expression of *Nr2f1*, another orphan nuclear receptor gene associated with neurodevelopmental disorders (*Bertacchi et al., 2020*; *Bosch et al., 2014*; *Contesse et al., 2019*), is high in the dorsal MP of the hippocampal primordium and is higher in the dorsal hippocampus than in the ventral hippocampus (*Figure 1*, *Figure 1—figure supplement 1*, and *Figure 5Ea*; *Flore et al., 2017*). Consistent with previous observations in *Emx1*^Cre/+; *Nr2f1*^F/F mutant mice (*Flore et al., 2017*), the dorsal hippocampus but not the ventral hippocampus was specifically shrunken in RX^Cre/+; *Nr2f1*^F/F mice (*Figure 2* and *Figure 5—figure supplement 1*). *Nr2f1* is expressed at the highest level in dorsal CA1 pyramidal neurons (*Figure 1*), indicating that the *Nr2f1* gene may play a role in the specification and differentiation of dorsal CA1 pyramidal neurons. As expected, the expression of Ctip2 and Wfs1, two markers for dorsal CA1 pyramidal neurons, could not be detected in the prospective dorsal CA1 domain in RX^Cre/+; *Nr2f1*^F/F mutant mice (*Figure 2*); furthermore, *Emx1*^Cre/+; *Nr2f1*^F/F mutant mice partially phenocopied the compromised development of the dorsal CA1 lineage (*Figure 2—figure supplement 1*). It seems that the spatiotemporal activity of RX-Cre recombinase is better or broader than that of the Emx1-Cre recombinase during the critical period of the specification of the dorsal CA1 pyramidal neuron identity. In addition, the Golgi staining assay revealed that the development of the dendrites of the dorsal CA1 pyramidal neurons was abnormal (*Figure 2—figure supplement 1*). All the observations above indicate that the *Nr2f1* gene is not only necessary for the morphogenesis of the dorsal hippocampus but is also required for the specification and differentiation of the dorsal CA1 pyramidal neurons, among which there are place cells. The identification of place cells 50 years ago was one of the most important breakthroughs in understanding the role of the hippocampus in memory (*O'Keefe and Dostrovsky, 1971*). Except for spatial information, place cells in the dorsal CA1 may also encode nonspatial representations, such as time (*Eichenbaum, 2017*; *Lisman et al., 2017*). Notably, 95% of patients carrying *Nr2f1* mutations are associated with ID. Here, our observations support the notion that the *Nr2f1* gene is a novel intrinsic regulator that specifies the dorsal CA1 pyramidal cell identity, which will benefit the understanding of both neurophysiological functions of the hippocampus and the etiology of NDD including ID and ASD.

## *Nr2f1* and *Nr2f2* cooperate to ensure the appropriate morphogenesis of the hippocampus by regulating the *Lhx5-Lhx2* axis in mice

In wild-type mice, *Nr2f1* and *Nr2f2* genes generated complementary expression profiles in the embryonic hippocampal primordium with *Nr2f1* in the dorsal MP, marked in green; *Nr2f2* in the ventral CH, marked in red (*Figure 5Ea* and *Figure 1—figure supplement 1*); and in the postnatal hippocampus with high-NR2F1 expression in the dorsal, marked in green; and high-NR2F2 expression in the ventral,

marked in red (*Figures 1 and 5Ea*). These findings indicated that *Nr2f* genes may coordinate to regulate hippocampal development. Indeed, as discussed above, the loss of either *Nr2f1* or *Nr2f2* only leads to dysplasia of the dorsal hippocampus (*Flore et al., 2017*; *Figure 2* and *Figure 2—figure supplement 1*) or ventral hippocampus (*Figure 1*), respectively; intriguingly, while both genes are efficiently excised by RX-Cre in the hippocampal primordium (*Figure 1—figure supplement 1*), more severely shrunken hippocampi developed in the 3-week-old double knockout mice (*Figure 3*). The dosage-dependent severity of hippocampal abnormalities suggested that two nuclear receptor genes, *Nr2f1* and *Nr2f2* could cooperate with each other to execute an essential and intrinsic function in the development of the hippocampus.

It is known that both extrinsic signals and intrinsic factors participate in the regulation of the early development of the hippocampus. Notably, mutations of *Wnt3a* and *Lef1* eliminate the entire hippocampus (*Galceran et al., 2000*; *Lee et al., 2000*). Given that the expression of *Axin2*, *Lef1*, and *Tcf4*, three Wnt-responsive transcription factor genes, was not altered in the *Nr2f* double mutant (*Figure 5A*), it is unlikely that abnormal Wnt signaling is the cause of the compromised hippocampus. *Lhx5* is specifically expressed in the hippocampal primordium and is required for the morphogenesis of the hippocampus (*Zhao et al., 1999*). *Lhx2* is necessary for hippocampal development by repressing cortical hem fate (*Mangale et al., 2008*; *Monuki et al., 2001*). Agenesis of the hippocampus is observed in either *Lhx5* or *Lhx2* null mutant mice, and these genes particularly repress each other (*Hébert and Fishell, 2008*; *Mangale et al., 2008*; *Roy et al., 2014*; *Zhao et al., 1999*), indicating that the proper expression levels of *Lhx5* and *Lhx2* genes are critical to maintain the appropriate development of the hippocampus (*Figure 5Eb*). The transcriptional and protein expression levels of *Lhx5* but not *Lhx2* were first reduced in the hippocampal primordium of *Nr2f* double-mutant mice at E11.5; later, enhanced expression of the Lhx2 protein was detected in the hippocampal primordium of double-mutant mice at E13.5 and E14.5 (*Figure 5A and B*). Moreover, the number of Lhx5-positive Cajal–Retzius cells was clearly reduced in the double-mutant embryos at E11.5, E13.5, and E14.5; consistent with the observations in the *Lhx5* null mutant (*Li et al., 2021*; *Miquelajáuregui et al., 2010*), the generation of Sox2-positive hippocampal NPCs was not affected, but the development of Tbr2-positive IPCs and NeuroD1-positive newborn neurons was abnormal in *Nr2f* double-mutant mice (*Figure 5*). Thus, our findings reveal a novel intrinsic regulatory mechanism that *Nr2f1* and *Nr2f2*, two disease-associated nuclear receptor genes, may cooperate with each other to ensure proper hippocampal morphogenesis by regulating the *Lhx5-Lhx2* axis. Intriguingly, compared with the adult *Emx1*^Cre/+^; *Nr2f1*^F/F^ mutant mice, the hippocampus was much smaller in the adult *Emx1*^Cre/+^; *Nr2f1*^F/F^; *Nr2f2*^F/F^ double-gene mutant mice; nevertheless, both the dorsal and ventral hippocampus were readily detected in double-gene mutants with *Emx1*^Cre^ (our unpublished observations). In addition, the discrepancy between the shrunken dorsal hippocampus associated with the loss of *Nr2f1* and the duplicated CA domains of the ventral hippocampus associated with the deficiency of *Nr2f2* suggested that the regulatory network related to *Nr2f* genes during the early morphogenesis of the hippocampus could be much more complicated than suspected and should be investigated in our future study.

### *Nr2f* genes are imperative for the formation of the trisynaptic circuit

The hippocampus and entorhinal cortex (EC) are interconnected through various neural circuits to mediate the flow of the information associated with declarative memory (*Basu and Siegelbaum, 2015*). Both direct and indirect glutamatergic circuits are involved in the relay of information from the EC to the hippocampal CA1, and the trisynaptic pathway is the most well-characterized indirect circuit. The EC sends sensory signals from association cortices via the perforant path to the DG, then the DG granule cells send excitatory mossy fiber projections to CA3 pyramidal neurons, and CA3 pyramidal neurons project to CA1 via the Schaffer collaterals (*Lee et al., 2020*). Consistent with the high expression of NR2F1 in the dorsal hippocampus and NR2F2 in the ventral hippocampus (*Figure 1*), the dorsal trisynaptic circuit is specifically damaged in *Nr2f1* mutants, as is the ventral trisynaptic circuit in *Nr2f2* mutant mice. Moreover, the hippocampal trisynaptic circuit was almost completely absent in the double mutants (*Figure 4*, and our unpublished observations). The information transfer associated with the trisynaptic circuits should be abolished particularly in the dorsal and/ or the ventral hippocampus in the above corresponding genetic mouse models. Interestingly, *Nr2f1* is also required to specify the medial EC cell fate (*Feng et al., 2021*). Therefore, the impaired formation

and function of trisynaptic circuits could be caused by the abnormal development of CA1, CA3, DG, or EC lineages. Nonetheless, given that newborn granule neurons are continuously generated in the adult DG to integrate into the existing neural circuits essential for declarative memory (*Toda et al., 2019*; *Tuncdemir et al., 2019*) and that hippocampal adult neurogenesis was severely compromised in the *Nr2f* double mutant (*Figure 3—figure supplement 1*), we could not exclude the possibility that impaired adult neurogenesis may also contribute to the malformation and impaired function of the trisynaptic pathway in double mutants. We would like to investigate what sort of synaptic circuitry is compromised either physiologically or morphologically in the trisynaptic circuit of individual animal model in detail in future studies.

The hippocampus is heterogeneous along its dorsoventral axis, and either the dorsal or ventral hippocampus generates unique and distinguishable characteristics of gene expression and connectivity, which enable the hippocampus to execute an integrative function from the encoding and retrieval of certain declarative memory to adaptive behaviors. Lesions of the dorsal hippocampus, which are essential for the cognitive process of learning and memory, lead to amnesia and ID; while damage to the ventral hippocampus, which is central for emotion and affection, is highly associated with psychiatric disorders including depression, anxiety, and schizophrenia. Our findings in this study reveal novel intrinsic mechanisms by which two nuclear receptor genes, *Nr2f1* and *Nr2f2*, which are associated with NDD or CHD, converge to govern the differentiation and integration of the hippocampus along the dorsoventral axis morphologically and functionally. Furthermore, this study provides novel genetic model systems to investigate the crosstalk among the hippocampal complex in gene expression, morphogenesis, cell fate specification and differentiation, connectivity, functions of learning/memory and emotion/anxiety, adaptive behaviors, and the etiology of neurological diseases. Nevertheless, many enigmas, such as whether and how the abnormalities of either the dorsal or ventral hippocampus affect the characteristics of the other, remain unsolved. In addition to the excitatory lineages and circuits, interneurons and inhibitory circuits play vital roles in maintaining the plasticity and functions of the hippocampus. We also wonder whether and how defects in interneurons and inhibitory circuits could contribute to the compromised morphogenesis, connectivity, and functions of the hippocampus and the etiology of psychiatric and neurological conditions, including ID, ASD, depression, anxiety, and schizophrenia.

# Materials and methods
## Animals
*Nr2f1-floxed* (*Nr2f1^{F/F}*) mice, *Nr2f2-floxed* (*Nr2f2^{F/F}*) mice, *Emx1^{Cre}* mice, and RX^{Cre} mice (*Swindell et al., 2006*) (PMID: 16850473) used in the study were of the C57B6/129 mixed background. The noon of vaginal plug day was set as the embryonic day 0.5 (E0.5). Only male mice at age of 10 wk or older were used in the Morris water maze. For other experiments, both male and female mice were used. All animal protocols were approved by the Institutional Animal Care and Use Committee (IACUC) at the Shanghai Institute of Biochemistry and Cell Biology, Chinese Academy of Sciences (protocols: SIBCB-NAF-14-001-S308-001). All methods were performed in accordance with the relevant guidelines and regulations. Only the littermates were used for the comparison.

## Nissl staining
We used xylene to dewax paraffin sections, followed by rinsing with 100, 95, and 70% ethanol. The slides were stained in 0.1% Cresyl Violet solution for 25 min. Then the sections were washed quickly in the water and differentiated in 95% ethanol. We used 100% ethanol to dehydrate the slides, followed by rinsing with the xylene solution. Finally, the neutral resin medium was used to mount the slides.

## Immunohistochemical (IHC) staining
The paraffin sections were dewaxed and rehydrated as described above for Nissl staining. The slides were boiled in 1× antigen retrieval solution (DAKO) under microwave conditions for 15 min. After cooled to room temperature (RT), the slides were incubated with 3% $H_2O_2$ for 30 min. Then, the slides were treated with blocking buffer for 60 min at RT and then incubated with the primary antibody in the hybridization buffer (10× diluted blocking buffer) overnight (O/N) at 4°C. The next day, the tyramide signal amplification kit (TSA) (Invitrogen) was used according to the manufacturer's protocol.

After being incubated with 1%TSA blocking buffer, the sections were treated with a biotinylated secondary antibody for 60 min at RT. After being washed with 1× PBS three times, the slides were incubated with 1× HRP-conjugated streptavidin for 1 hr. Next, the tyramide working solution was prepared, including the 0.15‰ $H_2O_2$ in distilled water, the 100× diluted tyramide substrate solution (tyramide-488 or tyramide-594), and the amplification buffer. The sections were incubated with the working solution for 10 min. Then the slides were counterstained with DAPI and mounted with the antifade mounting medium (Southern Biotech) (*Feng et al., 2017*; *Zhang et al., 2020*). Finally, the sections were observed and images were captured with a digital fluorescence microscope (Zeiss).

The following primary antibodies were used in the study: mouse anti-NR2F1 (1:1000, R&D, Cat# PP-H8132-00), mouse anti-NR2F2 (1:2000, R&D, Cat# PP-H7147-00), rabbit anti- NR2F2 (1:2000, a gift from Dr. Zhenzhong Xu, Zhejiang University, China), rabbit anti-HuB (1:500, Abcam, Cat# ab204991), rat anti-Ctip2 (1:500, Abcam, Cat# ab18465), rabbit anti-Wfs1 (1:500, ProteinTech, Cat# 11558-1-AP), goat anti-Prox1 (1:500, R&D, Cat# AF2727), rabbit anti-Calretinin (1:500, Sigma, Cat# C7479), rabbit anti-Calbindin (1:500, Swant, Cat# CB38), mouse anti-SMI312 (1:200, Covance, Cat# SMI-312R), rabbit anti-Sox2 (1:500, Affinity BioReagents, Cat# PA1-16968), rat anti-Tbr2 (1:500, Thermo Fisher, Cat# 12-4875-82), goat anti-NeuroD1 (1:200, Santa Cruz, Cat# sc-1084), goat anti-Lhx2 (1:200, Santa Cruz, Cat# sc-19344), goat anti-Lhx5 (1:200, R&D, Cat# AF6290), goat anti-β-galactosidase (LacZ) (1:400, Biogenesis, Cat# 4600-1409), mouse anti-GFAP (1:500, Sigma, Cat# G3893), rabbit anti-Nestin (1:200, Santa Cruz, Cat# sc-20978), and goat anti-Dcx (1:500, Santa Cruz, Cat# sc-8066). The following secondary antibodies were used in the study: donkey anti-mouse IgG biotin-conjugated (1:400, Jackson ImmunoResearch, Cat# 715-065-150), donkey anti-rabbit IgG biotin-conjugated (1:400, Jackson ImmunoResearch, Cat# 711-065-152), donkey anti-goat IgG biotin-conjugated (1:400, Jackson ImmunoResearch, Cat# 705-066-147), and donkey anti-rat IgG biotin-conjugated (1:400, Jackson ImmunoResearch, Cat# 712-065-150).

## Western blotting

We homogenized the isolated dorsal and ventral hippocampus tissues from 1-month-old mice respectively in the RIPA buffer (Applygen) with protease inhibitor cocktail (Sigma) and phosphatase inhibitors (Invitrogen) and then centrifuged at the speed of 12,000 rpm for 30 min. We collected the supernatants and analyzed the total concentrations by the BCA kit (Applygen). Gradient SDS-PAGE gels were used to separate the same amounts of protein sample (40 µg/lane), and then the proteins were transferred to the PVDF membranes (Millipore). After being blocked by the 3% BSA (Sigma) for 2 hr, the membranes that contained proteins were incubated by primary antibodies at 4°C O/N. The membranes were rinsed with 1× PBST three times for 10 min and then treated with biotinylated secondary antibodies for 2 hr at RT. After washing with 1× PBST, membranes were treated with the HRP-conjugated streptavidin for 1 hr at RT. Finally, we used the chemiluminescence detection system (Tanon) to detect the bands. The density of the protein band was analyzed by the software ImageJ.

The primary antibodies were used in the experiment as below: mouse anti-NR2F1 (1:2000, R&D, Cat# PP-H8132-00), rabbit anti-NR2F2 (1:3000, a gift from Dr. Zhenzhong Xu, Zhejiang University, China), and mouse anti-GAPDH (1:1000, Santa Cruz, Cat# sc-32233). The following secondary antibodies were applied in the study, including goat anti-mouse IgG biotin-conjugated (1:1000, KPL, Cat# 16-18-06) and goat anti-rabbit IgG biotin-conjugated (1:1000, KPL, Cat# 16-15-06).

## Golgi staining

Deep anesthesia was performed before sacrificing the control and mutant mice, and the brains were immediately isolated. The FD rapid GolgiStain kit (FD NeuroTech) was used to process the brain tissue samples, which were immersed in an equal volume of immersion solution mixed with solutions A and B and stored in the dark for 2 wk at RT. At least 5 mL of immersion solution was used for each cubic meter of the tissue. To achieve the best results, the container of tissue was gently swirled from side to side twice a week during the incubating period. Afterward, the brain tissue was transferred to solution C in the dark at RT for at least 72 hr (up to 1 wk). Finally, the tissues were cut into 100-µm-thick slices with a cryostat at –20 to –22°C and transferred to gelatin-coated microscope slides containing solution C using a sample retriever. The slices were dried naturally at RT. The concrete staining procedure of the kit was followed using the manufactory's protocol. Then, the sections were rinsed twice with double-distilled water for 4 min each time. The slices were placed in a mixture of one volume of

solution D, one volume of solution E, and two volumes of double-distilled water for 10 min and were rinsed twice with distilled water for 4 min each time. The sections were later dehydrated in 50, 75, and 95% ethanol for 4 min, respectively. Next, the slices were dehydrated in 100% ethanol four times for 4 min each time. Finally, the sections were cleared in xylene and mounted with a neutral resin medium.

## Morris water maze

By recording the time spent by the mice swimming in the water tank and finding the escape platform hidden underwater, and the swimming trajectory, the Morris water maze test can objectively reflect the spatial learning and memory ability of the mice. The test was divided into the control and mutant group with at least eight mice in each group. We poured tap water into the water maze tank and added an appropriate amount of well-mixed, milky white food dye. The height of the liquid level was about 1 cm higher than the escape platform, and the water temperature was kept at about 25 ± 1°C. At the same time, four markers of different shapes were pasted on the four directions of the inner wall above the water tank to distinguish different directions. The Morris water maze test was divided into the training phase and the probe trial phase. The training phase lasted for 6 d, four times a day, and the interval between each training was about 30 min. During training, the mice were placed into the tank from the entry points of four different quadrants facing the inner wall, and their latency was recorded from the time they entered the water to the time they found a hidden underwater platform and stood on it. After the mouse found the platform, we let it stay on the platform for 10 s before removing it. If the mouse failed to discover the platform 60 s after entering the water, it was guided to find the platform and left to stay for 10 s. Each mouse was placed into the water tank from four water entry points and recorded as one training session. The probe trial was carried out on the seventh day, and the underwater platform was removed. Each experimental mouse was put into the water tank at the same water entry point and allowed to move for 60 s. The time that each mouse spent in the quadrant, where the platform was originally placed, was recorded.

## RNA isolation and quantitative real-time PCR

Total RNAs were prepared from the whole telencephalon of the control (n = 5) and double-mutant (n = 3) mice at E11.5, respectively, with the TRIzol Reagent (Invitrogen) by following the manufactory's protocol. Reverse-transcription PCR and real-time quantitative PCR assays were performed as described previously (*Tang et al., 2012*). A Student's *t*-test was used to compare the means of the relative mRNA levels between the control group and mutant group. Primer sequences are as follows:

*Axin2-f*, 5'-CTGCTGGTCAGGCAGGAG-3', *Axin2-r*, 5'-TGCCAGTTTCTTTGGCTCTT-3'; *Nr2f1-f*, 5'-CAAAGCCATCGTGCTATTCA-3', *Nr2f1-r*, 5'-CCTGCAGGCTTTCGATGT-3'; *Nr2f2-f*, 5'-CCTCAAAG TGGGCATGAGAC-3', *Nr2f2-r*, 5'-TGGGTAGGCTGGGTAGGAG-3'; *Emx1-f*, 5'-CTCTCCGAGACG CAGGTG-3', *Emx1-r*, 5'-CTCAGACTCCGGCCCTTC-3'; *Emx2-f*, 5'-CACGCTTTTGAGAAGAACCA-3', *Emx2-r*, 5'-GTTCTCCGGTTCTGAAACCA-3'; *Foxg1-f*, 5'-GAAGGCCTCCACAGAACG-3', *Foxg1-r*, 5'-GGCAAGGCATGTAGCAAAAG-3'; *Gli3-f*, 5'-TGATCCATCTCCTATTCCTCCA-3', *Gli3-r*, 5'-TCTGGATA CGTCGGGCTACT-3'; *Lef1-f*, 5'-TCCTGAAATCCCCACCTTCT-3', *Lef1-r*, 5'-TGGGATAAACAGGCTG ACCT-3'; *Lhx2-f*, 5'-CAGCTTGCGCAAAAGACC-3', *Lhx2-r*, 5'-TAAAAGGTTGCGCCTGAACT-3'; *Lhx5-f*, 5'-TGTGCAATAAGCAGCTATCCA-3', *Lhx5-r*, 5'-CAAACTGCGGTCCGTACA-3'; *Otx1-f*, 5'-CCAG AGTCCAGAGTCCAGGT-3', *Otx1-r*, 5'-CCGGGTTTTCGTTCCATT-3'; *Otx2-f*, 5'-GGTATGGACTTG CTGCATCC-3', *Otx2-r*, 5'-CGAGCTGTGCCCTAGTAAATG-3'; *Pax6-f*, 5'-GTTCCCTGTCCTGTGGACTC -3', *Pax6-r*, 5'-ACCGCCCTTGGTTAAAGTCT-3'; *Tcf4-f*, 5'-AAATGGCCACTGCTTGATGT-3', *Tcf4-r*, 5'-GCACCACCGGTACTTTGTTC-3'.

## Quantification and statistical analysis

The number of specified immunofluorescent marker-positive cells was assessed by ImageJ Cell Counter in full image fields. Three brain sections per mouse were counted for each index. GraphPad Prism 7.0 (GraphPad) was used to perform statistical analysis. The data analysis used one-way ANOVA, Dunnett's or Tukey's post hoc tests, and Student's unpaired *t*-test. The data are expressed as the mean ± SEM. The data obtained from at least three independent replicates were used for statistical analysis. p<0.05 was considered the significant statistical difference.

## Acknowledgements

We thank Ms. Emerald Tang for her assistance with the manuscript. This work was supported by the National Natural Science Foundation of China (31671508) and Guangdong Provincial Basic and Applied Basic Research Fund (2021A1515011299) to KT; and in part by the National Key Basic Research and Development Program of China (2018YFA0108500, 2019YFA0801402, 2018YFA0800100, 2018YFA0108000, 2018YFA0107200, 2017YFA0102700), 'Strategic Priority Research Program' of the Chinese Academy of Sciences, Grant No. (XDA16020501, XDA16020404) to NJ.

## Additional information

### Funding

| Funder | Grant reference number | Author |
|---|---|---|
| National Natural Science Foundation of China | 31671508 | Ke Tang |
| Guangdong Provincial Basic and Applied Basic Research Fund | 2021A1515011299 | Ke Tang |
| National Key Basic Research and Development Program of China | 2018YFA0108500 | Naihe Jing |
| National Key Basic Research and Development Program of China | 2019YFA0801402 | Naihe Jing |
| National Key Basic Research and Development Program of China | 2018YFA0800100 | Naihe Jing |
| National Key Basic Research and Development Program of China | 2018YFA0108000 | Naihe Jing |
| National Key Basic Research and Development Program of China | 2018YFA0107200 | Naihe Jing |
| National Key Basic Research and Development Program of China | 2017YFA0102700 | Naihe Jing |
| Chinese Academy of Sciences | Strategic Priority Research Program XDA16020501 | Naihe Jing |
| Chinese Academy of Sciences | Strategic Priority Research Program XDA16020404 | Naihe Jing |

The funders had no role in study design, data collection and interpretation, or the decision to submit the work for publication.

### Author contributions

Xiong Yang, Designed and conducted the experiments, Organized and wrote the manuscript; Rong Wan, Designed and conducted the experiments; Zhiwen Liu, Assisted in conducting the experiments and analyzed the data; Su Feng, Assisted in conducting the experiments and analyzed the data; Jiaxin Yang, Assisted in conducting the experiments and analyzed the data; Naihe Jing, Conceived the project and approved the manuscript; Ke Tang, Organized and wrote the manuscript, Conceived the project and approved the manuscript

## Author ORCIDs

Naihe Jing (ORCID) https://orcid.org/0000-0003-1509-6378
Ke Tang (ORCID) https://orcid.org/0000-0002-0516-4667

## Ethics

All animal protocols were approved by the Animal Ethics Committee of the Shanghai Institute of Biochemistry and Cell Biology. All methods were performed in accordance with the relevant guidelines and regulations.(Protocols: SIBCB-NAF-14-001-S308-001).

Reviewer #1 (Public Review): https://doi.org/10.7554/eLife.86940.3.sa1
Reviewer #2 (Public Review): https://doi.org/10.7554/eLife.86940.3.sa2
Reviewer #3 (Public Review): https://doi.org/10.7554/eLife.86940.3.sa3
Author Response https://doi.org/10.7554/eLife.86940.3.sa4

## Additional files

### Supplementary files

• MDAR checklist

### Data availability

Numerical data are available in the manuscript and supporting files.

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
