## [Editor Report · eLife assessment]

This is an **important** study demonstrating distinct roles for the nuclear receptor genes COUP-TFI and COUP-TFII in hippocampal development. The strength of evidence is **compelling**, using rigorous state-of-the-art methods to demonstrate functional redundancy of these genes in regulating the Lhx2/Lhx5 axis. The major strengths of the study are the dramatic morphogenic phenotypes, and the resultant altered gene networks. These findings have theoretical or practical implications beyond a single field, and will be of interest to geneticists, developmental neurobiologists and chromatin biologists among others.

---

## [Referee Report · Reviewer #1 (Public Review)]

The hippocampus is a structure in the cerebral cortex known to be compartmentalised into regions with different functions. Dorsal hippocampus is involved in cognitive functions such as declarative memory and spatial navigation and interconnects chiefly with the neocortex. Ventral hippocampus interconnects with limbic structures such as amygdala and hypothalamus and is involved in affective states and anxiety. What specifies this functional regionalisation during development is not well understood. The present study focuses on the role of transcription factors COUPTFI and COUPTFII, confirming a previously observed dorsal to ventral gradient of expression of COUPTFI in both embryonic and adult mouse hippocampus, and reporting that expression of COUPTFII is strongest in ventral hippocampus. The aim of the authors was then to probe the role of these transcription factors with the use of conditional knockout of one or both factors using RxCre+ mice (sometimes Emx1Cre+ for comparison). As predicted, COUPTFI insufficiency resulted in failure of the CA1 subregion of the dorsal hippocampus to develop properly (with concomitant loss of performance in a spatial memory task) COUPTFII knockdown had even more marked effects upon the ventral hippocampus with ectopic CA1/CA3 domains forming, while a double knockout lead to a drastic reduction in size of the hippocampus with subsequent effects upon the appearance of hippocampal synaptic circuitry and the capacity for adult neurogenesis (a feature of rodent hippocampus). In order to help explain the role of COUPTFI/II a role in regulating expression of two transcription factors LHX2 and LHX5, known to be crucial to hippocampal development, was tested by examining gene and protein expression. Changes in LHX2 and LHX5 was observed and a role for COUPTFI/II in regulating expression of these genes was postulated.

I believe the authors have largely achieved their aims and the results mostly support the conclusions, but, as discussed further below, there are some weaknesses in the data and some areas that could be expanded upon and improved. The methods are mostly appropriate. The use of the transgenic mice and the application of histological methods, especially tyramide amplified immunohistochemistry, is exemplary. However, I'm not sure a wide enough range of tests to explore the phenotype of the transgenic mice was employed to back the conclusions drawn by the authors. The introduction and discussion are nicely written and explain the general concepts and conclusions well. The work makes an important contribution to our understanding of brain development in general and hippocampal development in particular.

Turning to more specific comments, I must first point out that specification of the ventral hippocampus by expression of COUPTFII is not an entirely original finding, as it was suggested for the developing human hippocampus following immunohistochemical experiments illustrating COUPTFII expression to be confined to the ventral hippocampal structures of the medial temporal cortex (doi: 10.1093/cercor/bhx185). Of course, this study, unlike the present study, was restricted to fetal cortex, not adult, and also reported expression of COUP-TFI throughout dorsal and ventral hippocampal structures but without observing any dorsal to ventral gradient, however I feel its contribution to the field has been overlooked by the present study, and should be incorporated into the introduction and/or discussion.

More information about Rx-cre mice would be informative and could help explain the different phenotype observed when EMX1-cre mice were used to conditionally knock down COUPTFI/II expression.

The demonstration of antagonistic gradients of COUP-TFI and -TFII across the hippocampus is more convincing in the immunohistochemical preparations than in the western blots. The qualitative data presented in Fig.1p does not convincingly represent the quantitative data presented in Fig.1q. There seem to be multiple bands for COUP-TFII and I wonder exactly how quantifying this was approached?

Behavioural testing is limited to one test of dorsal hippocampus function. other tests for non-spatial memory, e.g. novel object recognition, or ventral hippocampus function, e.g. step through passive avoidance, might have lead to some interesting discriminations between the various knock down animals (see doi: 10.3389/fnagi.2018.00091).

Abnormalities in the trisynaptic circuit. No studies of actual synapses, either physiological or morphological, were carried out. I wonder to what extent these immunohistochemical studies just further reflect the abnormalities in hippocampal morphology presented earlier in the manuscript without specifically telling us about synaptic circuits? Although the immunohistochemical preparations are beautiful, they are inadequate on their own in telling us much about what sort of synaptic circuitry exists in the transgenic animals.

LHX2/LHX5 interaction. The immunohistochemical study, which shows clear differences in LHX5 and LHX2 protein expression at E14.5 in double knockdown mice is more convincing than the qPCR study at E11.5, which show surprisingly small differences in mRNA expression. Could the authors expand upon whether this is due to stage of development, or differences between mRNA and protein expression? Why hasn't both mRNA and protein expression data at both time points been presented?

Response to the re-submission

I am happy that the western blot presentation has been improved, and my minor comments attended to. It is disappointing, although understandable given the timeframe, that the lack of qPCR data at 14.5 ED has not been rectified. The immunohistochemical data alone is qualitative and only indicative of LHX5 expression remaining depressed and LHX2 expression possibly increasing between E11.5 and E14.5. In the absence of qPCR data, a more quantitative immunohistochemical study, such as counting blind the number of LHX5+ Cajal-Retzius cells, or measuring optical density of LHX2 expression under rigorous experimental conditions regarding image collection and processing, would be required to support the hypothesis that COUPTFI/II expression modulates the LHX2/LHX5 axis.

---

## [Referee Report · Reviewer #2 (Public Review)]

The Author's chose to limit their response to re-doing the Lhx5 immuno using the correct antibody which now displays the expected staining: Lhx5 expression is limited to the hem. They have not however presented a characterization of where the RxCre acts, although this was pointed out by other reviewers as well. It would have been useful to demonstrate the expression domain in particular with respect to the time of its initiation, to explain how it causes a phenotype close to that described for the Lhx5 knockout (Zhao et al., 1999). From the decrease of Lhx5 expression and the CR cells which arise from the hem, it appears that the RxCre does indeed act in the hem. However, the timing and spatial pattern is important to establish, as I had pointed out in my first review, "If [the expression of RxCre] it has a dorso-ventral bias in the early embryo, it could explain the regional difference in the COUPTF phenotypes."

The major interpretive criticisms I made have not been addressed even though these would have only required a re-writing and re-interpretation of the data. The revised manuscript continues to include major errors of interpretation such as the idea that Lhx2 and Lhx5 "inhibit each other", something that is unsupported since the expression domains of these two genes are mutually exclusive as is clear from the authors' own new data and the literature.Lines 355-360: "The expression of Lhx2 was comparable between the control and double-mutant mice at E11.5 (Figure 5Be-h, e'-h'). Interestingly, the expression of the Lhx2 protein was increased in the hippocampal primordium in the COUP-TF double-mutant mice at E13.5 and E14.5 (Figure 5Bm-p, m'-p', u-x, u'-x'). The upregulation of Lhx2 expression is most likely associated with the reduced expression of the Lhx5 gene"There's clearly no Lhx5 in the hippocampal primordium so how is this possible?

The authors have missed the insights from key papers that they cite, e.g. (lines 352-354) " The expression of Lhx2 was expanded ventrally into the choroid plexus in the Lhx5 null mutant mice (Zhao et al., 1999)" - this paper in fact shows there is no choroid plexus. Lhx2 appears to extend to the midline likely because the hem isn't specified. The authors would benefit from reading https://doi.org/10.1101/2022.10.25.513532 in which Lmx1a is shown to be the master regulator of the hem.

A sentence like (lines 77-81) further blurs the literature: "Intriguingly, deficiency of either Lhx5 or Lhx2 results in agenesis of the hippocampus, and more particularly, these genes inhibit each other (Hébert & Fishell, 2008; Mangale et al., 2008; Roy, Gonzalez-Gomez, Pierani, Meyer, & Tole, 2014; Zhao et al., 1999), indicating that the Lhx5 and Lhx2 genes may generate an essential regulatory axis to ensure the appropriate hippocampal development"

First, none of the papers they cite shows that these two factors inhibit each other. Second, the "agenesis of the hippocampus" in the Lhx2 knockout mentioned in Porter et al. (1997) was later shown to be due to a transformation of the hippocampal primordium into an EXPANDED hem (Mangale et al.) In contrast, the "agenesis of the hippocampus" in the Lhx5 mutant appears to be due to the near-complete LOSS of the hem and evidenced by the loss of its derivatives, the choroid plexus and the CR cells (Zhao et al., 1999). The fact that loss of these two factors have opposite effects on the hem (each resulting in loss of the hippocampus, one due to transformation of the hippocampal primordium into hem and the other because of a lack of hipopcampal induction) does not mean that there is an Lhx5-Lhx2 "axis" regulating hippocampal development.

I won't repeat my other comments here, but the majority of them were not addressed in any way.

In conclusion, I find it unfortunate that the authors have chosen not to use the detailed input provided by the reviewers which would have greatly improved their manuscript.

---

## [Referee Report · Reviewer #3 (Public Review)]

The authors have made significant improvements in addressing my major concerns raised during the previous review. However, I still have some lingering concerns regarding the quantification and statistical analysis presented in the manuscript. Specifically, there is a lack of robust quantification and statistical analysis to support the conclusions drawn, particularly in relation to the numbers of DG, CA1, and CA3 neurons.

To strengthen the validity and reliability of the findings, I would strongly recommend the authors to incorporate a more rigorous quantitative approach in their research. This could involve implementing stereological methods or other appropriate techniques to accurately estimate the numbers of neurons in the DG, CA1, and CA3 regions. By doing so, the authors would enhance the credibility of their conclusions and provide more solid evidence for their claims.

---

## [Author Response]

The following is the authors’ response to the current reviews.

We will make some minor changes to address the issues in the revised manuscript during preparation of the Version of Record.

1. Acknowledge the previous discovery that COUPTFII expression is confined to the ventral hippocampus in early human fetal forebrain (doi: 10.1093/cercor/bhx185).

We agree. We will incorporate the previous discovery that COUPTFII expression is confined to the ventral hippocampus in early human fetal forebrain (doi: 10.1093/cercor/bhx185) in the discussion section of "COUP-TFII governs the distinct characteristics of the ventral hippocampus".

1. Give some consideration to this observation from my original review "Abnormalities in the trisynaptic circuit. No studies of actual synapses, either physiological or morphological, were carried out. I wonder to what extent these immunohistochemical studies just further reflect the abnormalities in hippocampal morphology presented earlier in the manuscript without specifically telling us about synaptic circuits? Although the immunohistochemical preparations are beautiful, they are inadequate on their own in telling us much about what sort of synaptic circuitry exists in the transgenic animals".

Our data in Figure 4 show clearly that at the neural circuit level, compared with the corresponding control, the trisynaptic circuit is abnormal in all three models; therefore, in the discussion section of "COUP-TF genes are imperative for the formation of the trisynaptic circuit", we will add the following sentence, "We would like to investigate what sort of synaptic circuitry is compromised either physiologically or morphologically in the trisynaptic circuit of individual animal model in detail in the future studies.

In addition, we will correct a reference related to the COUP-TFII gene and congenital heart defects.

The reference of "High, F. A., Bhayani, P., Wilson, J. M., Bult, C. J., Donahoe, P. K., & Longoni, M. (2016). De novo frameshift mutation in COUP-TFII (NR2F2) in human congenital diaphragmatic hernia. Am J Med Genet A, 170(9), 2457-2461. doi:10.1002/ajmg.a.37830" was replaced with "Al Turki, S., Manickaraj, A. K., Mercer, C. L., Gerety, S. S., Hitz, M. P., Lindsay, S., . . . Hurles, M. E. (2014). Rare variants in NR2F2 cause congenital heart defects in humans. Am J Hum Genet, 94(4), 574-585. doi:10.1016/j.ajhg.2014.03.007".

—————

The following is the authors’ response to the original reviews.

**Reviewer #1(Recommendations For The Authors):**
1. Better presentation of the western blot results

We agree with the reviewer. Based on the suggestion, new information about the western blot results has been added in the revised Figure 1Ap. We added a dash to each western blot image to indicate the target band of COUP-TFI (46 KDa), COUP-TFII (45 KDa), and GAPDH (37 KDa), respectively. There were two bands in the blot of COUP-TFII, with the upper band corresponding to mouse IgG at 50 KDa, and the bottom band corresponding to COUP-TFII protein at 45 KDa. Therefore, only the lower bands of COUP-TFII are used for the quantitative analysis. The expression of COUP-TFII in the ventral hippocampus is clearly higher than that in the dorsal hippocampus.

1. Full presentation of the Immunohistochemistry and qPCR results for at E11.5 and E14.5 in double knockdown mice.

Thanks for the suggestion. Based on the suggestion, we added immunofluorescent data in the double knockout mice at E11.5 in the Figure 5Ba-h. Meanwhile, given that it takes time to prepare animal samples at E14.5 for RT-qPCR assays, we performed immunofluorescent assays at both E13.5 and E14.5 to make sure that the changes of Lhx5 and Lhx2 expression in the hippocampal regions between the control and mutant mice were consistent. As shown in the new Figure 5B, consistent with the downregulated expression of Lhx5 transcripts in the double mutant, the expression of the Lhx5 protein was reduced in the CH in the double mutants at E11.5; moreover, the numbers of Lhx5-positive Cajal-Retzius cells decreased in the double mutant embryos at E11.5, E13.5 and E14.5 (Figure 5Ba-d, a’-d’, a’’-d’’, i-l, i’-l’, q-t, q’-t’). Consistent with RT-qPCR data, the expression of Lhx2 was comparable between the control and double-mutant mice at E11.5 (Figure 5Be-h, e’-h’). Interestingly, the expression of the Lhx2 protein was increased in the hippocampal primordium in the COUP-TF double-mutant mice at E13.5 and E14.5 (Figure 5Bm-p, m’-p’, u-x, u’-x’). Please find the altered descriptions in the Page 15, lines 347-351, 353-358 and Page 21, lines 500-503 in the revised manuscript.

1. Minor corrections. Lines 159-162, prospected not quite the right word. I would suggest "an ectopic CA-like region was observed medially in the temporal hippocampus in the COUP1TFII mutant, where the prospective posterior part of the medial amygdaloid nucleus was situated, (MeP), indicated by the star (Figure 1Ba-f). The presence of the ectopic CA-like region in the ventral but not dorsal hippocampus of the mutant was further confirmed by the presence of the prospective MeP and amygdalohippocampal area (AHi) in sagittal sections, as indicated by the star. See also line 251. Line437/438 I would suggest "... most important breakthroughs in understanding the role of the hippocampus in memory."

Thanks for the suggestion. We made the changes based on the suggestion. Please find the amendments in Page 8, lines 178-181; Page 12, lines 270, 276; Page 14, line 318; Page 19, lines 451; Page 20, lines 461-462 in the revised manuscript.

**Reviewer #2 (Recommendations For The Authors):**
1. It is also important to point out that the immunofluorescence data in Figure 5B is contrary to what is known for Lhx5 (it's not expressed in the neocortical and hippocampal vz) and Lhx2 (it's not expressed in the choroid plexus). Authors should explain how their conclusions could align more clearly, and consider the possibility that their results are due to a possible artifact of image setting issues or worse, antibody specificity issues.

Very good point. Based on the comments and suggestions, we first tested another Lhx5 antibody, R&D, Cat# AF6290, in the immunofluorescence assays. Indeed, there was something wrong with the previous Lhx5 antibody, Millipore, Cat# AB5762. With the new Lhx5 antibody, consistent with the reported in situ data, the expression of Lhx5 was detected specifically in the CH at E11.5, and in the Cajal-Retzius cells in the marginal zone of the telencephalon. The same Lhx2 antibody, Santa Cruz, Cat# sc-19344, which has been used successfully in one of our previous studies (Tang et al., Development, 2012) (PMID: 22492355), was used in the present study. We believe that the observations at the MP and DP of the samples are really associated with the expression of Lhx2 protein. We performed new immunofluorescence assays with the new Lhx5 antibody and confirmed with the Lhx2 antibody. As shown in new Figure 5B, consistent with the downregulated expression of Lhx5 transcripts in the double mutant, the expression of the Lhx5 protein was reduced in the CH in the double mutants at E11.5; moreover, the numbers of Lhx5-positive Cajal-Retzius cells decreased in the double mutant embryos at E11.5, E13.5 and E14.5 (Figure 5Ba-d, a’-d’, a’’-d’’, i-l, i’-l’, q-t, q’-t’). Consistent with RT-qPCR data, the expression of Lhx2 was comparable between the control and double-mutant mice at E11.5 (Figure 5Be-h, e’-h’). Interestingly, the expression of the Lhx2 protein was increased in the hippocampal primordium in the COUP-TF double-mutant mice at E13.5 and E14.5 (Figure 5Bm-p, m’-p’, u-x, u’-x’). Please find the changed descriptions in Page 15, lines 347-351, 353-358 and Page 21, lines 500-503 in the revised manuscript.

The reference:

Tang, K., Rubenstein, J. L., Tsai, S. Y., & Tsai, M. J. (2012). COUP-TFII controls amygdala patterning by regulating neuropilin expression. Development, 139(9), 1630-1639. doi:10.1242/dev.075564

1. The expression domain of RxCre remains poorly explained, and the early expression of COUPTFI and II (E10.5-E12.5) could be considered major weaknesses of the paper.

Thanks for the suggestion. The generation of RXCre was reported by Swindell et al., Genesis, 2006 (PMID: 16850473). Given that the activation of the LacZ expression serves as an indicator for the deletion of the COUP-TFII gene (Tang et al., Development, 2012) (PMID: 22492355), we performed the immunofluorescent data with antibodies against COUP-TFII and LacZ on the sagittal sections of RXCre/+; COUP-TFIIF/+ heterozygous mutant and RXCre/+; COUP-TFIIF/F homozygous mice at E11.5. As shown in the new Figure 1—figure supplement 1Da-f, COUP-TFII was readily detected at the hippocampal primordium of the heterozygous mutant embryo at E11.5 (Figure 1—figure supplement 1Da, c, g); in contrast, the expression of COUP-TFII significantly decreased in the homozygous mutant (Figure 1—figure supplement 1Dd, f, j). In addition, compared with the heterozygous mutant embryo, the LacZ signals increased distinctly in the hippocampal primordium of the homozygous mutant embryo at E11.5 (Figure 1—figure supplement 1Db-c, e-f, h, k), suggesting that RXCre recombinase can efficiently excise the COUP-TFII gene in the hippocampal primordium as early as E11.5. Please find the corresponding changes in Page 7, lines 149-159 and Page 8, lines 160-164 in the revised manuscript.

Meanwhile, we also added the early expression of COUP-TFI and -TFII at E10.5 and E11.5 in new Figure 1—figure supplement 1Aa-d. At embryonic days 10.5 (E10.5), COUP-TFI was detected in the dorsal pallium (DP) laterally and COUP-TFII was expressed in the MP and CH medially (Figure 1—figure supplement 1Aa, b). At E11.5, the expression of COUP-TFII remained in the hippocampal primordium, including MP and CH (Figure 1—figure supplement 1Ac, d). Please find the corresponding changes in Page 6, lines 129-132 and Page 9, lines 202-203 in the revised manuscript.

The references:

Swindell, E. C., Bailey, T. J., Loosli, F., Liu, C., Amaya-Manzanares, F., Mahon, K. A., . . . Jamrich, M. (2006). Rx-Cre, a tool for inactivation of gene expression in the developing retina. Genesis, 44(8), 361-363. doi:10.1002/dvg.20225

Tang, K., Rubenstein, J. L., Tsai, S. Y., & Tsai, M. J. (2012). COUP-TFII controls amygdala patterning by regulating neuropilin expression. Development, 139(9), 1630-1639. doi:10.1242/dev.075564

**Reviewer #3 (Recommendations For The Authors):**
1. Regarding the RxCre line, I was also confused about its spatiotemporal expression, as this line is not a commonly used Cre line and no detailed description is provided in the manuscript. Searching this line shows a previous paper by the authors (PMID: 22492355) in which they tested the RxCre recombinase activity. At E12.5, RxCre induced high LacZ expression in the ventral telencephalon but much less in the dorsal telencephalon. But they did not check later stage. Therefore, it's hard to explain the defective dorsal hippocampus in RxCre, CFI CKO. They should check later stage.

The generation of RXCre was reported by Swindell et al., Genesis, 2006 (PMID: 16850473), which reveals high Cre recombinase activity of RXCre in the eye and ventral telencephalon. Given that the activation of the LacZ expression serves as an indicator for the deletion of COUP-TFII gene, Tang et al., Development, 2012 (PMID: 22492355), we performed the immunofluorescent data with antibodies against COUP-TFII and LacZ on the sagittal sections of RXCre/+; COUP-TFIIF/+ heterozygous mutant and RXCre/+; COUP-TFIIF/F homozygous mice at E11.5. As shown in new Figure 1—figure supplement 1D, compared with the heterozygous mutant embryo, the expression of COUP-TFII was significantly decreased in the homozygous mutant; in addition, the LacZ signals evidently increased in the hippocampal primordium of the homozygous mutant embryo at E11.5, suggesting that RXCre recombinase can efficiently excise the target gene in the hippocampal primordium as early as E11.5. The expression of COUP-TFI is barely detectable in the early developing hippocampal primordium including MP at E10.5, E11.5 and E12.5. The expression of COUP-TFI is high in the MP of the control (Figure 1Cj, l); in contrast, the COUP-TFI expression is barely detectable in the MP of the homozygous double mutant at E14.5, indicating that RXCre can efficiently delete the COUP-TFI gene in the hippocampal primordium at E14.5. The loss of the COUP-TFI gene in the MP as early as E14.5 by RXCre initiates the defective dorsal hippocampus in RXCre/+; COUP-TFIF/F knockout mice.

1. Authors should check and review extensively for improvements to the use of English.

We carefully checked and made changes throughout the manuscript accordingly. For example, “imperative” was used 6 times in the previous manuscript, lines 20, 255, 486, 499, 522, 553; “imperative” was used only once in Page 22, line 522 in the revised manuscript.

1. Please correct the manuscript; 1-month-old mice are not adult mice.

Thanks for the suggestion. Based on the suggestion, we have corrected related words and sentences in the manuscript. Please find the amendments in the revised manuscript (Page 7, line 146; Page 9, lines 203-204; Page 10, line 213; Page 13, lines 299-300; Page 17, line 406; Page 20, line 476).

1. Additional ref should be added at line 93 on page 5.

Based on the suggestion, we added some new references (Bertacchi et al., EMBO J, 2020) (PMID: 32572460); (Del Pino et al., Cereb Cortex, 2020) (PMID: 32484994); (J. Feng et al., Sci Adv, 2021) (PMID: 34215582) at line 96 on page 5.

The references:

Bertacchi, M., Romano, A. L., Loubat, A., Tran Mau-Them, F., Willems, M., Faivre, L., . . . Studer, M. (2020). NR2F1 regulates regional progenitor dynamics in the mouse neocortex and cortical gyrification in BBSOAS patients. Embo j, 39(13), e104163. doi:10.15252/embj.2019104163

Del Pino, I., Tocco, C., Magrinelli, E., Marcantoni, A., Ferraguto, C., Tomagra, G., . . . Studer, M. (2020). COUP-TFI/Nr2f1 Orchestrates Intrinsic Neuronal Activity during Development of the Somatosensory Cortex. Cereb Cortex, 30(11), 5667-5685. doi:10.1093/cercor/bhaa137

Feng, J., Hsu, W. H., Patterson, D., Tseng, C. S., Hsing, H. W., Zhuang, Z. H., . . . Chou, S. J. (2021). COUP-TFI specifies the medial entorhinal cortex identity and induces differential cell adhesion to determine the integrity of its boundary with neocortex. Sci Adv, 7(27). doi:10.1126/sciadv.abf6808

1. I am confused why the authors analyzed 1-month-old mice in some instances but 3-month-old mice in others.

The RXCre/+; COUP-TFIF/F; COUP-TFIIF/F double mutant mice barely survived beyond postnatal 3 weeks. To make our findings consistent and comparable, we mainly prepared figures with observations on about 1-month-old mice in the RXCre related single or/and double gene mutant mouse models. In the study of the Emx1Cre related COUP-TFI mouse model, due to behavioral tests such as the Morris water maze test, experiments were performed with the adult experimental animal about postnatal 3 months. In order to be consistent with the stage of the mice for the behavioral tests, we only displayed morphological data with observations on the control and Emx1Cre/+; COUP-TFIF/F mutant mice at about postnatal 3-month.